# Enhanced sulfur in the UTLS in spring 2020

Laura Tomsche[1,2], Andreas Marsing[1,2], Tina Jurkat-Witschas[1], Johannes Lucke[1,5], Stefan Kaufmann[1], Katharina Kaiser[3], Johannes Schneider[3], Monika Scheibe[1], Hans Schlager[1], Lenard Röder[3], Horst Fischer[3], Florian Obersteiner[4], Andreas Zahn[4], Martin Zöger[6], Jos Lelieveld[3], Christiane Voigt[1,2]

[1]Deutsches Zentrum für Luft- und Raumfahrt (DLR), Institute of Atmospheric Physics, 82234 Oberpfaffenhofen, Germany
[2]Johannes Gutenberg University Mainz, 55099 Mainz, Germany
[3]Max Planck Institute for Chemistry, 55128 Mainz, Germany
[4]Karlsruhe Institute for Technology (KIT), 76021 Karlsruhe, Germany
[5]Faculty of Aerospace Engineering, Delft University of Technology, 2629 Delft, Netherlands
[6]Deutsches Zentrum für Luft- und Raumfahrt (DLR), Flight Experiments, 82234 Oberpfaffenhofen, Germany

*Correspondence to*: Laura Tomsche (ltomsche@uni-mainz.de)

**Abstract.** Sulfur compounds in the upper troposphere and lower stratosphere (UTLS) impact the atmosphere radiation budget, either directly as particles or indirectly as precursor gas for new particle formation. In situ measurements in the UTLS are rare, but are important to better understand the impact of the sulfur budget on climate. The BLUESKY mission in May/June 2020 explored an unprecedented situation. 1) The UTLS experienced extraordinary dry conditions in spring 2020 over Europe, in comparison to previous years and 2) the first lockdown of the COVID-19 pandemic caused major emission reductions from industry, ground, and airborne transportation. With the two research aircraft HALO and Falcon, 20 flights were conducted over Central Europe and the North Atlantic to investigate the atmospheric composition with respect to trace gases, aerosol, and clouds. Here, we focus on measurements of sulfur dioxide ($SO_2$) and particulate sulfate ($SO_4^{2-}$) in the altitude range of 8 to 14.5 km which show unexpectedly enhanced mixing ratios of $SO_2$ in the upper troposphere and of $SO_4^{2-}$ in the lowermost stratosphere. In the UT, we find $SO_2$ mixing ratios of $(0.07 \pm 0.01)$ ppb, caused by the remaining air traffic, reduced $SO_2$ sinks due to low OH and low cloud fractions, and to a minor extend by uplift from boundary layer sources. Particulate sulfate showed elevated mixing ratios of up to 0.33 ppb in the LS. We suggest that the eruption of the volcano Raikoke in June 2019, which emitted about 1 Tg $SO_2$ into the stratosphere in northern midlatitudes caused these enhancements, in addition to Siberian and Canadian wildfires and other minor volcanic eruptions. Our measurements can help to test models and lead to new insights in the distribution of sulfur compounds in the UTLS, their sources and sinks. Moreover, these results can contribute to improve simulations of the radiation budget in the UTLS with respect to sulfur effects.

## 1 Introduction

The stratospheric aerosol layer changes in time, especially after volcanic eruptions with plume injection heights into the stratosphere, the layer gets more pronounced (Kremser et al., 2016). It plays a role in the radiative balance and thus impacts

the climate (Solomon et al., 2011). An enhanced aerosol concentration leads to a larger albedo. The geoengineering community investigates the enhancement of the aerosol layer with injections of sulfur compounds into the stratosphere to partly counteract greenhouse gases related global warming (Crutzen, 2006; Schäfer et al., 2015). The stratospheric aerosol layer is often referred

to as "Junge layer" and can extend from the tropopause up to 25 km (Junge et al., 1961). The chemical composition of the stratospheric aerosol layer is dominated by sulfate ($SO_4^{2-}$) particles, which consist mainly of pure sulfuric acid droplets, sulfuric acid with material from ablated meteoroids or mixed organic-sulfate particles (Murphy et al., 2014; Cziczo et al., 2001; Schneider et al., 2021). During volcanic quiescent periods, precursor gases, like carbonyl sulfide (OCS), and non-volcanic sulfur dioxide ($SO_2$), as well as tropospheric $SO_4^{2-}$ particles preserve the stratospheric layer (Brock et al., 1995). Due to its

long lifetime, OCS is vertically uplifted from the tropics into the stratosphere and there it converts mostly through photodissociation to $SO_2$ (Sheng et al., 2015). The $SO_2$ chemistry and transport depends strongly on the ambient conditions. In the free troposphere and lower stratosphere, $SO_2$ reacts predominantly with hydroxyl (OH) to sulfuric acid ($H_2SO_4$) (English et al., 2011; Stockwell and Calvert, 1983), thus the lifetime correlates with the OH concentration. At cold temperatures and in the presence of water vapour, the gaseous $H_2SO_4$ condenses quickly to particles (Almeida et al., 2013; Kirkby et al., 2011),

thereby forming sulfate aerosol. In the boundary layer, pollution could significantly reduce the lifetime to hours (Lee et al., 2011) and consequently, also the transport of $SO_2$ to higher altitudes. Clouds could also limit the $SO_2$ lifetime to hours or days (Lelieveld et al., 1993), as the conversion of $SO_2$ to $H_2SO_4$ is faster in cloud droplets than in the gas phase (Seinfeld and Pandis, 2006). Nevertheless, $SO_2$ can be transported from the planetary boundary layer (PBL) into the UTLS region via different pathways. Similar to OCS, $SO_2$ can be vertically transported across the tropical tropopause layer (TTL) or by overshooting

convection in the tropics (Fueglistaler et al., 2009) or by the transport of $SO_2$ in a warm conveyor belt (WCB) in the midlatitudes (Arnold et al., 1997; Clarisse et al., 2011; Fiedler et al., 2009) or by transport processes connected with the Asian monsoon (Gottschaldt et al., 2017, 2018; Ploeger et al., 2017; Tomsche et al., 2019; Vogel et al., 2019; von Hobe et al., 2021). An explosive volcanic eruption can inject huge amounts of ash, $SO_2$, and other volcanic gases into the stratosphere and thus enhance the stratospheric aerosol layer (Kremser et al., 2016). In 1991, the volcano Mount Pinatubo (15°N) emitted

approximately 20 Tg of $SO_2$ and 30 Tg of aerosol (McCormick et al., 1995), which impacted the stratosphere globally. Volcanic eruptions in midlatitudes can also impact the stratosphere, e.g. Mount St Helens (46°N, 0.8 Tg $SO_2$) in 1980, but its impact vanished in about a year (Deshler et al., 2006). One recent midlatitude eruption of similar strength was the volcano Raikoke in June 2019 (48.28°N, Kloss et al., 2021, de Leeuw et al., 2021) which emitted approx. 1.4-1.6 Tg $SO_2$. A further important source of stratospheric aerosol is intense wildfires, which can potentially develop pyrocumulonimbus (pyroCb) and

thus transport biomass burning emissions into the UTLS (Fromm et al., 2005; Peterson et al., 2018). Moreover, air traffic is another source of particles and precursor gases in the UTLS (Lee et al., 2010; Voigt et al., 2010; Jurkat et al., 2011).
In late spring 2020, the UTLS region was probed over Europe during the BLUESKY mission, the period covered the first weeks of the coronavirus disease 2019 (COVID-19) lockdown in Europe, which caused reductions in emissions from industry, ground, and especially airborne transportation (Voigt et al., 2022). Under these conditions, we found enhanced values of $SO_2$

in the upper troposphere (UT) and of particulate $SO_4^{2-}$ in the lowermost stratosphere (LS), which motivated us to investigate these sulfur compounds with respect to their sources and sinks.

In the following, we first present the BLUESKY mission in section 2. In section 3.1, we introduce the airborne measurements and in section 3.2 the trace gas and particulate profiles obtained during BLUESKY and also show tropospheric and stratospheric influenced compounds in tracer-tracer correlations (section 3.3). Afterwards, we focus on the $SO_2$ profile in the

UT (section 4) and continue in section 5 with the stratospheric sulfate aerosol. Finally, we summarize our results and give an outlook in section 6.

## 2 Methods

### 2.1 BLUESKY mission

The BLUESKY mission was led by the German Aerospace Center (DLR) and the Max-Planck-Institute for Chemistry, Mainz.

Coordinated flights were performed from Oberpfaffenhofen with the High-Altitude and Long-Range Research Aircraft HALO and the DLR Falcon over Europe and the North Atlantic between 16 May and 09 June 2020. In total 20 flights were performed (Fig. 1). The period covered the first weeks of the COVID-19 lockdown in Europe and thus offered a unique opportunity to investigate an unprecedented situation of reduced anthropogenic emissions from industry, ground, and airborne transportation. The goal of BLUESKY was to explore the changes in the atmospheric composition and gain new insights on how

anthropogenic emissions perturb chemical and physical processes in the atmosphere. Both aircraft were equipped with in situ instruments to investigate trace gases, aerosols, and cloud properties. The payload of both aircraft was complementary to obtain a comprehensive dataset. Especially, during five days with coordinated flights over Germany (23 May, 26 May, and 28 May) and over the North Atlantic (30 May, 02 June), the payload offered the opportunity to probe the air masses in more detail. An overview of the BLUESKY mission is given in Voigt et al. (2022) and further detailed studies are published by Schumann et

al. (2021a, 2021b), Reifenberg et al. (2022), Nussbaumer et al. (2022), Hamryszczak et al. (2022) and Krüger et al. (2022). In the present study we will focus on the sulfur compounds in the upper troposphere and lower stratosphere.

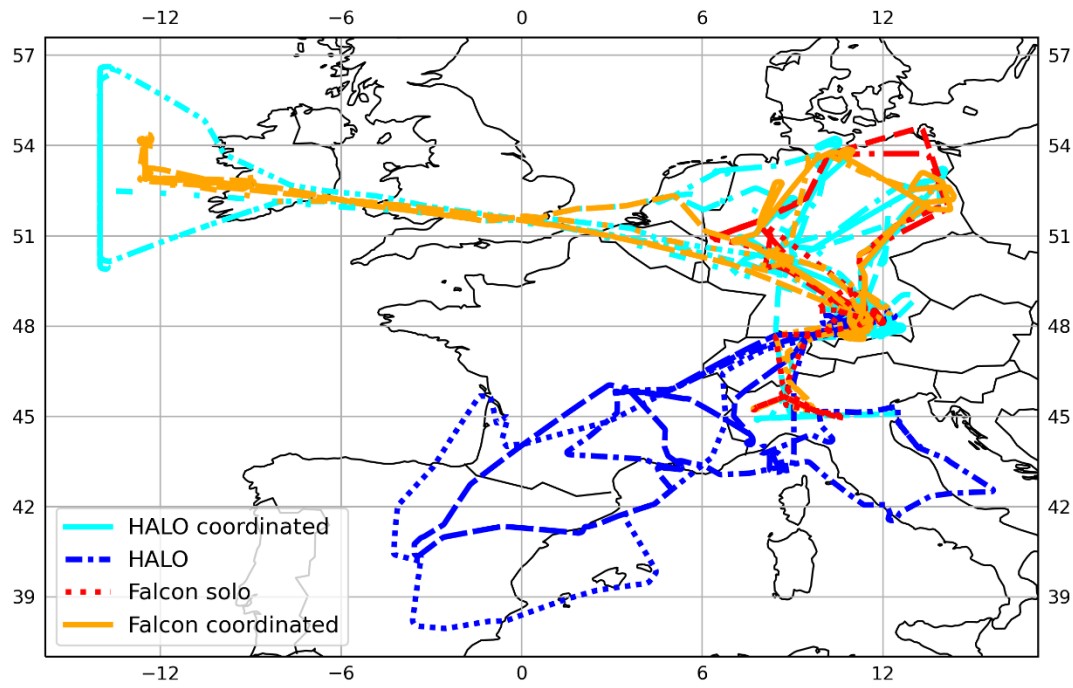

**Figure 1: Overview of all flight tracks performed by Falcon (red) and HALO (blue) during the BLUESKY mission in May/June 2020 during the COVID-19 lockdown. Coordinated flights (Falcon: orange; HALO: cyan) were performed over Germany on 23 May, 26 May, and 28 May and twice (30 May and 02 June) as both aircraft headed towards the North Atlantic, west of Ireland.**

## 2.2 Instrumentation

In the present study several trace gas measurements onboard Falcon and also trace gas and particle measurements onboard HALO are used. Onboard Falcon, the atmospheric chemical ionization mass spectrometer AIMS measures gaseous $SO_2$ and nitric acid ($HNO_3$) among other compounds at mixing ratios relevant for the UTLS region by using $SF_5^-$ as reagent ion. A more detailed description of the instrument can be found elsewhere (Voigt et al., 2014; Jurkat et al., 2016; Marsing et al., 2019). $SO_2$ is calibrated in-flight using an isotopically labelled calibration gas mixture of the isotope $^{34}SO_2$, which is heavier than the naturally dominant isotope $^{32}SO_2$, but has the same chemical behaviour (Jurkat et al., 2016). The natural isotopic ratio is $^{34}S/^{32}S = 0.0454$ and the mass spectrometer can detect both isotopes separately as they differ in mass by 2 amu (atomic mass units). This has the advantage that the calibration gas can be continuously added to the sampling flow and the system is well conditioned for $SO_2$. A drawback is that the background of the instrument is increased by 5%, due to impurities of $^{32}SO_2$ in the calibration gas. The $SO_2$ data are instrument background corrected, which includes a moisture

correction, as higher water vapour concentrations lead to cross sensitivities on $m/z = 83$ amu ($FSO_2^-$; Jurkat et al., 2016).
With increasing moisture in lower altitudes, a correction is more difficult and reduces the data quality. As the focus of the present study is the UTLS region, we limited our analysis on altitudes above 8 km and thus ensure the data quality. The other trace gas measured by AIMS is $HNO_3$, which is in-flight calibrated using a permeation oven with a solution of $HNO_3$ in water (Jurkat et al., 2014). The data are background corrected including a moisture correction, which is necessary to account for cross sensitivities caused by water vapour (Jurkat et al., 2016). The AIMS measurements were performed with a 1.6 sec
time resolution and smoothed with a running mean of 20 seconds. The 1 σ detection limit of $SO_2$ varied between 0.006-0.017 ppb. The total uncertainty is on average 22.7% for $SO_2$ and included the uncertainty of the moisture correction. The 1-σ detection limit for $HNO_3$ is in the range of 0.005-0.009 ppb. The $HNO_3$ total uncertainty is on average 16% (Marsing, 2021). Further measurements onboard Falcon included CO and $O_3$. $O_3$ was measured using a UV photometer (Schulte and Schlager, 1996; Ziereis et al., 2000), CO was measured by cavity ring down spectroscopy (Klausner et al., 2020). The
accuracies for CO and $O_3$ are 15% and 5%, respectively. Additionally, water vapor ($H_2O$) was measured with the lyman-alpha absorption instrument integrated in the meteorological sensor system.

Onboard HALO, the compact time-of-flight aerosol mass spectrometer (C-ToF-AMS) measured the aerosol composition (Drewnick et al., 2005; Schmale et al., 2010; Schulz et al., 2018). Aerosol particles of approximately 50 to 800 nm are analysed, which then provides quantitative mass concentrations of organic matter, sulfate, nitrate and ammonium. The instrument is
equipped with a constant pressure inlet that ensures a steady mass flow and an operation pressure of the aerodynamic lens for stable inflight operation (Molleker et al., 2020). Here, we focus on sulfate. For a better comparability of $SO_4^{2-}$ with $SO_2$, we calculate mixing ratios (ppb) instead of using concentrations (µg m$^{-3}$). Additionally, mixing ratios have the advantage of being pressure independent. We assumed that all $SO_4^{2-}$ would be evaporated and calculated the volumetric mixing ratio for $SO_4^{2-}$. Above 8 km, the 1 σ detection limit is $(0.006 \pm 0.001)$ ppb for $SO_4^{2-}$, the accuracy is 30% and the precision on average
$(0.002 \pm 0.001)$ ppb (Schulz et al., 2018). Additionally, the trace gases CO and $O_3$ onboard HALO are considered in the present study for altitudes above 8 km. CO was measured by the TRacer In-Situ Tdlas for Atmospheric Research (TRISTAR; Schiller et al., 2008) with a total uncertainty of 3 % for tropospheric measurements (Nussbaumer et al., 2021). Note that due to a small nitrous oxide ($N_2O$) interference the uncertainty in the lower stratosphere is higher ($8.5 \pm 3.9$ ppbv). $O_3$ was measured by the Fast Airborne Ozone instrument FAIRO, which combines the technique of a UV photometer and a chemiluminescence detector
(Zahn et al., 2012) and the total uncertainty of $O_3$ was 2-2.5%. Water vapor was measured with the tunable diode laser (TDL) hygrometer SHARC (Sophisticated Hygrometer for Atmospheric ResearCh).
SO$_2$ was sampled onboard Falcon and $SO_4^{2-}$ was probed onboard HALO. Nevertheless, campaign averaged CO and $O_3$ profiles from both aircraft agree and motivate the combined interpretation of the $SO_2$ and $SO_4^{2-}$ distributions during spring 2020 (see Sec. 3.2).

## 2.3 Trajectory calculations

Back trajectory calculations were performed using the HYSPLIT atmospheric transport and dispersion model (Stein et al., 2015; Rolph et al., 2017) with the GDAS (Global Data Assimilation System) meteorological dataset (Kanamitsu, 1989). For selected cases with either elevated $SO_2$ or $SO_4^{2-}$ mixing ratios, 360 h back trajectory ensembles were calculated. One ensemble consists of 27 single trajectories, which are offset by one meteorological grid point in the horizontal and 0.01 sigma in the vertical coordinate. With the help of the back trajectories, air mass origins and transport pathways in the atmosphere could be identified.

## 3 Results

### 3.1 Trace gases along flight track

On 02 June 2020, HALO and Falcon took off in Oberpfaffenhofen and headed towards the North Atlantic, west of Ireland, with similar flight tracks, altitude and similar take-off time. With a shorter flight range, Falcon landed in Shannon/ Ireland to refuel, while HALO continued its flight. In Figure 2, flight altitude and in situ measurements are plotted against longitude for comparability from both aircraft for the time between 7:00 and 10:00 UTC. $O_3$ and CO were measured aboard both aircraft. Both trace gases show similar patterns independent of the aircraft. The highest $O_3$ mixing ratios with maxima of 475 ppb (Falcon) and 423 ppb (HALO) were approximately between 6°W and 3°E, while CO had the lowest mixing ratios of 18 ppb (Falcon) and 28 ppb (HALO) and vice versa along the longitudes outside this range, when CO mixing ratios were enhanced with maxima of 110 ppb (Falcon) and 109 ppb (HALO), $O_3$ mixing ratios were low (Falcon: 74 ppb, HALO: 62 ppb). Because CO and $O_3$ from both platforms agree within their uncertainties and reflect the same trends, we assume that both aircraft probed the same air mass. Between 6°W and 3°E, $O_3$ and $HNO_3$ are positively correlated, as expected, and $HNO_3$ mixing ratios increase up to 1.6 ppb. A similar trend can be observed in the particulate compound $SO_4^{2-}$, which also is enhanced when $O_3$, as a stratospheric tracer, is enhanced. $SO_4^{2-}$ ranges from the detection limit to 0.21 ppb. $SO_2$ mixing ratios show a larger variability, but in general, they follow the CO mixing ratios, which is a tropospheric tracer (Fischer et al., 2000; Hoor et al., 2002). $SO_2$ ranges from the detection limit to 0.15 ppb and experiences one short peak with 0.15 ppb around 6.5°W at an altitude of 11.9 km. Other trace gases onboard Falcon measured 266 ppb $O_3$, 70 ppb CO, and 0.8 ppb $HNO_3$ during this event, but beside $HNO_3$, which increased slightly, CO and $O_3$ showed no perturbations around this location. An in-depth analysis and discussion on potential explanations for this or similar features is presented in sections 4.

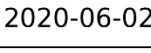

**Figure 2: On 02 June 2020, HALO and Falcon had similar flight tracks from Oberpfaffenhofen towards the North Atlantic, west of Ireland. The sampling was roughly between 7-10 UTC. Plotted are in a) $SO_2$, b) $SO_4^{2-}$ c) CO, d) $O_3$, e) $HNO_3$ and f) altitude across longitudes 15°W-12°E.**

## 3.2 SO₂ and SO₄²⁻ median profiles

Following the case mentioned above, we broaden our analysis to all flights of the whole campaign. In Figure 3, median, 25[th] and 75[th] percentile profiles of the trace gases and the particulate compounds are displayed with the potential temperature as a vertical axis. The medians, 25[th] and 75[th] percentiles are calculated for 5 K potential temperature bins from 310 K to 355 K for Falcon flights and up to 385 K for HALO flights. The Falcon profile is limited in height due to the maximum flight altitude of approx. 12.5 km in comparison to HALO with a ceiling altitude of 14.5 km. First, we compare the trace gases measured on both aircraft, Falcon and HALO. The stratospheric tracer $O_3$ behaves similar for both aircraft within the 25[th] and 75[th] percentiles with a step around 340 K, while the spread between the percentiles starts to increase already around 330 K. Below this altitude, median $O_3$ mixing ratios reach minima of 41 ppb (Falcon) and 62 ppb (HALO) and above they rise up to 420 ppb (Falcon) and 642 ppb (HALO). The chemical tropopause is marked by strong gradients in several tracers. $O_3$ as a common indicator shows a kink in its median profile at around 140 ppb and 340 K potential temperature, which is within the limits given by Thouret et al. (2006). The dynamical tropopause, displayed as the 2 PVU based on ECMWF/ERA5 data along the Falcon flight tracks (Fig. 3), is around 335 K and thus has a similar height as the chemical tropopause. The tropospheric tracer CO from both platforms has similar profiles with median mixing ratios of 37-112 ppb (Falcon) and 12-96 ppb (HALO). Both CO profiles show a decrease with height, thus anticorrelated to $O_3$, but they reflect a significant change in the mixing ratios at 340 K, similar to $O_3$. The profiles of the tropospheric tracer $H_2O$ also decrease with height, following the CO profile, but with a less pronounced step around the chemical tropopause. The $H_2O$ mixing ratios are 26-268 ppm (Falcon) and 3-153 ppm (HALO). Additionally, the median $HNO_3$ profile follows the trend of $O_3$ with low mixing ratios down to 0.3 ppb followed by a steep increase around 340K and reaches a maximum of 1.4 ppb. Between 330 K and 350 K, the stratospheric tracers $O_3$ and $HNO_3$, as well as the tropospheric tracer CO show larger variations between the percentiles, which indicates the mixing layer described by Hoor et al. (2002) over the course of the mission. As shown, the profiles for $O_3$ and CO for HALO and Falcon flight tracks agree well within their 25[th] and 75[th] percentiles. Both aircraft probed the atmosphere above Europe and the North Atlantic within the mission period and thus during similar meteorological conditions, even though the days and routes partly differ. This gives us confidence to further investigate and compare measurements of sulfur compounds sampled on both aircraft: $SO_2$ was sampled on Falcon, while $SO_4^{2-}$ was probed on HALO. Profiles of $SO_2$ and $SO_4^{2-}$ are shown in Fig. 3a and 3b, respectively. The median $SO_2$ profile decreases with height and median values range from 0.05 ppb to 0.08 ppb with the lowest mixing ratios above 330 K. The opposite behaviour can be observed in the median $SO_4^{2-}$ profile, which increases with height. The mixing ratios are lowest at 0.02 ppb

and raise up to 0.33 ppb. The profile shows a similar trend in comparison to $O_3$ with a stepwise increase around 340 K. The enhanced $SO_4^{2-}$ mixing ratios above the chemical tropopause can be associated with stratospheric sulfate aerosol.

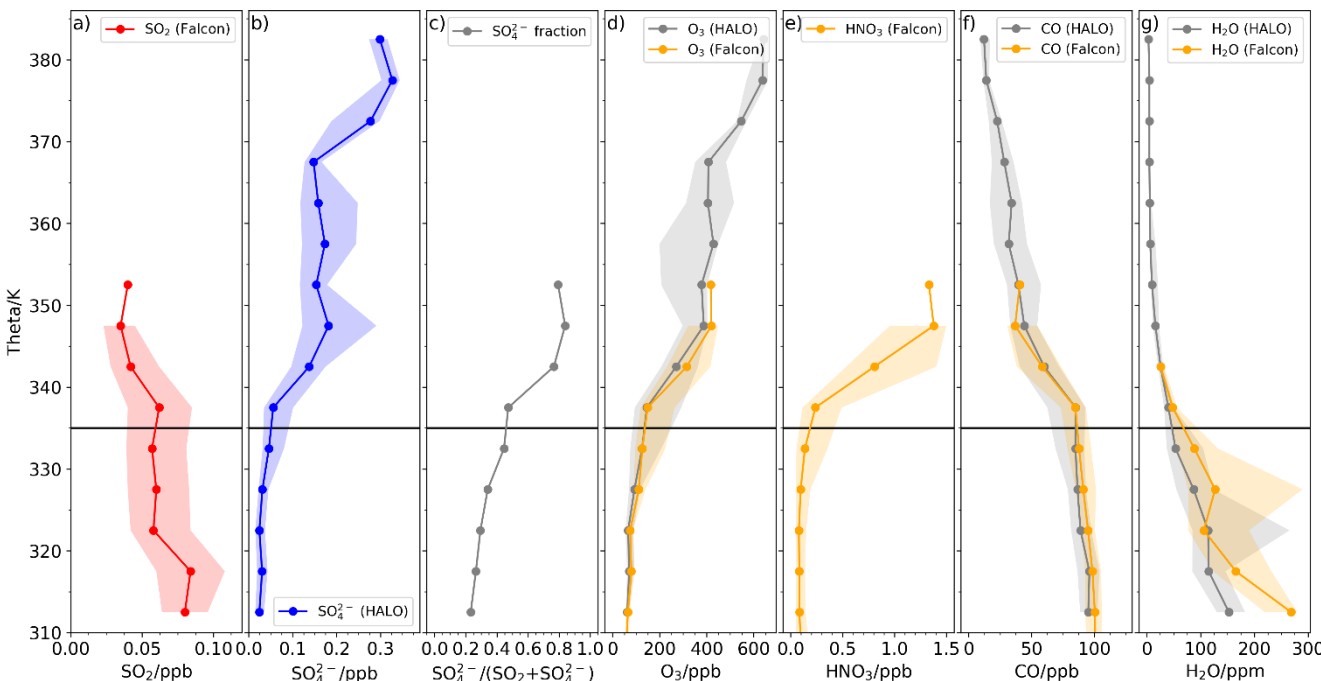

**Figure 3: Median profiles with 25th and 75th percentiles as shaded areas are shown for trace gases with the potential temperature as vertical axes. The data are calculated for 5K potential temperature bins. In: a) $SO_2$, b) $SO_4^{2-}$, c) the $SO_4^{2-}$ / ($SO_2$+ $SO_4^{2-}$) ratio d) $O_3$, e) $HNO_3$, f) CO, and g) $H_2O$ for measurements performed on HALO and Falcon. Additionally, the black line at 335 K roughly indicates the dynamical tropopause (as 2PVU) based on ECMWF/ERA5 analysis along the Falcon flight tracks.**

In Figure 3c the profile of the ratio of $SO_4^{2-}$/($SO_2$+ $SO_4^{2-}$) is plotted. The ratio is a measure of the relative contribution from the precursor $SO_2$ to the total sulfur budget. Below the chemical tropopause most of the $SO_2$ is still present as $SO_2$, while above the tropopause $SO_4^{2-}$ dominates the ratio. The sum of $SO_2$ and $SO_4^{2-}$ for the median profiles is rather stable around 0.10 ppb between 310 K and 340 K; above the chemical tropopause the sum increases up to on average 0.20 ppb and is dominated by the enhancement of $SO_4^{2-}$ in the range 340-355 K. In the next section we will discuss potential sources explaining the distribution of sulfur compounds in the UTLS.

**3.3 Stratospheric and tropospheric influenced air masses**

In order to obtain an overview on the distribution of sulfur compounds with respect to the chemical tropopause, Fig. 4 shows tracer-tracer correlations of $O_3$, CO, and $HNO_3$, comparable to previous studies investigating the cross-tropopause exchange

and the chemical composition of the tropopause (Fischer et al., 2000; Hoor et al., 2002). In contrast to the median profiles in the previous section, here all available data are plotted, either for HALO (considering $SO_4^{2-}$) or for Falcon (considering $SO_2$).

In the correlation plot between $O_3$ and CO color-coded with $SO_4^{2-}$ (Fig. 4a) from HALO for the whole altitude range (0-14 km), the stratospheric branch is visible with low CO and high $O_3$ values, while the tropospheric branch is characterised by low $O_3$ and high CO values, similar to e.g. Fischer et al., (2000). The transition layer is clearly visible, the layer is a mixing layer, influenced by air masses with stratospheric and tropospheric origin (Hoor et al., 2002). Without the exchange processes across the tropopause, we would expect a L-shape profile (Fischer at al., 2000). The mixing layer almost extends

over the whole $O_3$ range from 150 ppb to 400 ppb, similar to other mixing layers in the same latitude and season (Hoor et al., 2002; Pan et al., 2004) and thicker in comparison to a winter polar mixing layer (Fischer et al., 2000). The higher $SO_4^{2-}$ mixing ratios are either in the (unmixed) stratospheric branch or partly mixed into in the upper part of the transition layer. With respect to the chemical tropopause, the elevated $SO_4^{2-}$ mixing ratios appear only in the stratosphere ($O_3 \geq 120$ ppb; Thouret et al., 2006). In Figure 4b) the correlation between $O_3$ and CO with and without color-coded $SO_2$ onboard Falcon is

displayed. A subset of the Falcon flights is missing there due to missing $O_3$ data in the beginning of the campaign. As the Falcon mainly operated up to 12 km, the pure stratospheric branch is hard to identify, while the tropospheric branch is clearly identifiable with the black dots (without $SO_2$). However, within the mixing layer, the stratospheric and tropospheric influences still differ, which is reflected in the $SO_2$ mixing ratios. In order to cover all Falcon flights, we use here $HNO_3$ as a stratospheric tracer. In Figure 4c, the $HNO_3$ to CO correlation for the measurements onboard Falcon are plotted with color-

coded $SO_2$, the Figure includes all Falcon flights, Figure 4b and 4c show similar patterns for $SO_2$, with higher mixing ratios towards more tropospheric influence and lower mixing ratios when the stratospheric influence dominates. One $SO_2$ outlier with higher mixing ratios at enhanced $HNO_3$ und reduced CO can be identified (Fig. 4c; grey circle). A possible explanation could be (aged) aircraft plume encounters (as observed e.g. in Jurkat et al., 2011), as HYSPILT back trajectory calculations tend towards long range transport in the UTLS region for this case (Fig. 5a/b). Generally, the trajectories do not indicate

transport from local PBL sources for cases with elevated $SO_2$. In Figure 5a-h, examples for cases with elevated $SO_2$ are shown, including the case on 02 June 2020 described in section 3.1 (Fig. 5g/h). Additionally, Hamryszczak et al. (2022) found that hydrogen peroxides were scavenged by clouds during BLUESKY in the lower and middle troposphere (0-7 km). As $SO_2$ can easily be scavenged by clouds (Seinfeld and Pandis, 2006), the potential for $SO_2$ being transported from the local PBL to the UT seems unlikely. The tracer-tracer-correlation and the relative distribution of $SO_2$ along the transition layer

show no direct link to the $SO_4^{2-}$ distribution, as they are in different regimes, thus we assume that they originate from different sources. The back trajectories of $SO_2$ and $SO_4^{2-}$ cases support the assumption that the origins differ, especially concerning the different altitude ranges of the trajectories (Fig. 5).

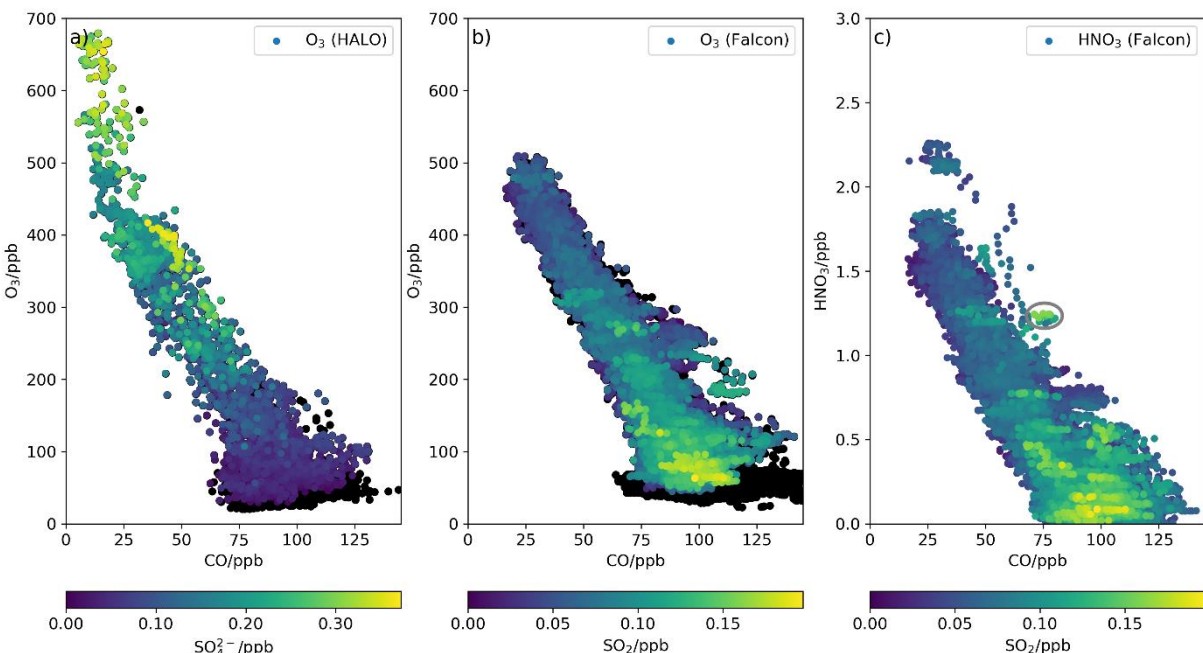

Figure 4: Tracer-tracer correlation for a) 30 sec data CO - $O_3$ in black for whole altitude range and for heights above 8 km $SO_4^{2-}$ is color-coded from HALO flights, b) CO - $O_3$ in black for whole altitude range, and with color-coded $SO_2$ (above 8 km) from Falcon flight, when $O_3$ was available, and c) 1 sec data CO - $HNO_3$ with color-coded $SO_2$ from Falcon flights. In c) a grey circle marks an outlier with high $SO_2$, CO and $HNO_3$.

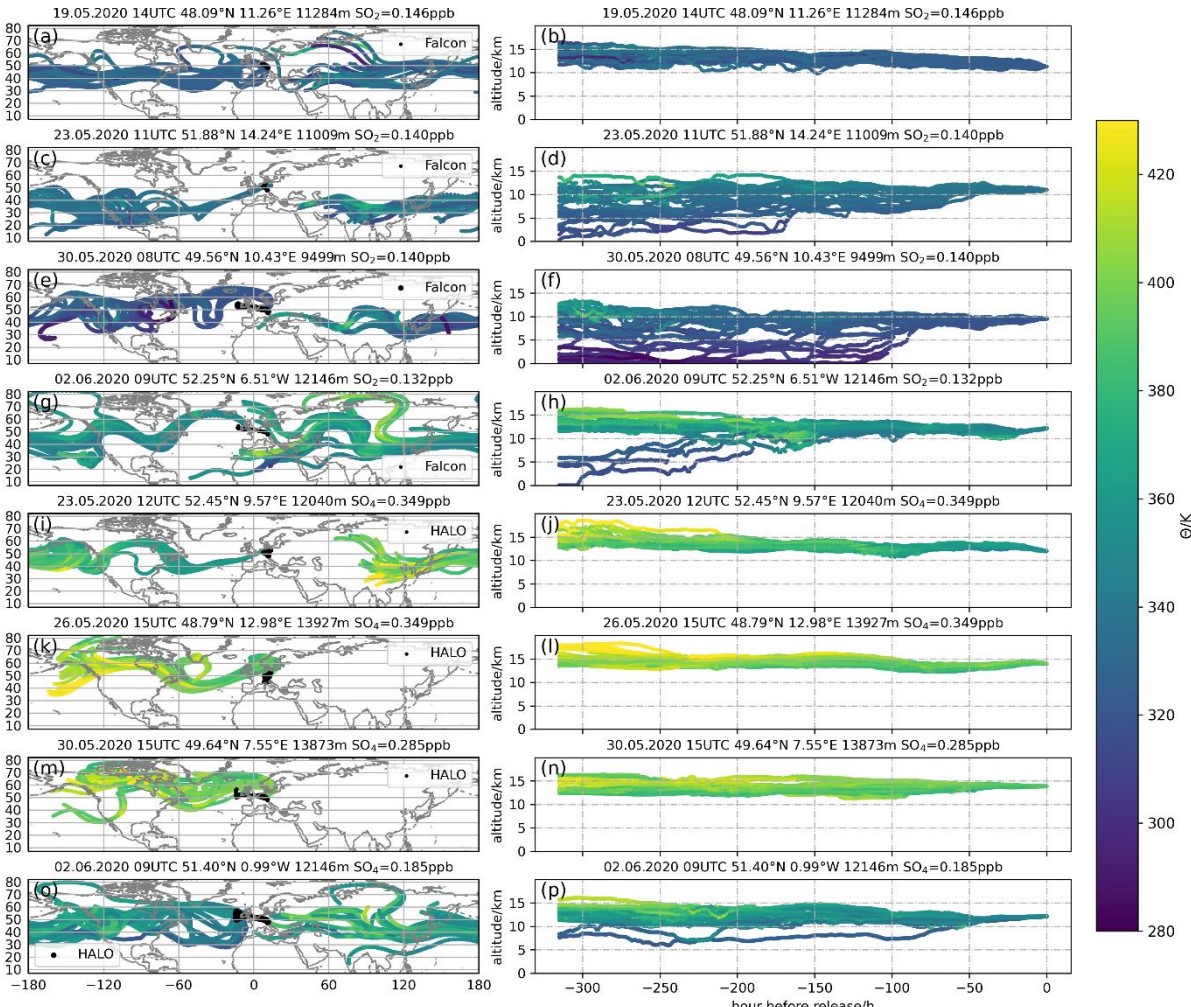

**Figure 5: HYSPLIT 360 hours back trajectories calculated for cases with elevated SO₂ (a-h; Falcon) and SO₄²⁻ (i-p; HALO) mixing ratios. The release points started in the vicinity of these events. In the left column: in black are the flight tracks, color-coded is the potential temperature along the trajectories to indicate the transport altitude. The right column represents the trajectories as hours before release vs. altitude, also with color-coded potential temperature The cases of enhanced SO₂ were on 19 May 2020 (a, b), 23 May 2020 (c, d), 30 May 2020 (e,f), and 02 June 2020 (g, h). The cases of elevated SO₄²⁻ were on 23 May 2020 (i, j), 26 May 2020 (k, l), 30 May 2020 (m, n) and 02 June 2020 (o, p).**

## 4 Enhanced SO$_2$ in the UT

In Figure 6a, the median SO$_2$ profile with shaded 25[th] and 75[th] percentile is plotted against the flight altitude, in order to compare the data with literature values. The SO$_2$ percentiles are calculated for 500 m bins. Similar to Fig. 3b, the median SO$_2$ decreases with height from around 0.08 ppb to 0.04 ppb. Previous in situ SO$_2$ measurements at similar flight altitudes are included in Fig. 6a. Overall, the BLUESKY SO$_2$ measurements are within the range of previous airborne studies, even though all studies are snapshots of the atmosphere of different locations on the Northern hemisphere, different seasons, and different meteorological situations. Keeping this in mind, we will have a closer look. Williamson et al. (2021) reported low mixing ratios for the remote Northern hemispheric background over the Pacific for the upper troposphere as well as for the lowermost stratosphere during the ATom mission (2016-2018). Speidel et al. (2007) reported higher SO$_2$ values for the upper tropospheric background over Europe and the eastern Atlantic in summer 2004. Jurkat et al. (2010) measured the stratospheric background over Europe in autumn 2008 in a similar range to Speidel et al. (2007). Above 12 km the BLUESKY SO$_2$ mixing ratios agree well with the stratospheric background from Jurkat et al. (2010) and the tropospheric background from Speidel et al. (2007). Surprisingly, the BLUESKY SO$_2$ profile slightly exceeds the previous measurements below these altitudes in the upper troposphere. The upper tropospheric SO$_2$ profile compares better to SO$_2$ mixing ratios, which were associated with the SO$_2$ background in the North Atlantic flight corridor in 1997 or 2010 (Arnold et al., 1997; Jurkat et al., 2010). Arnold et al. (1997) reported SO$_2$ mixing ratios in the range of 0.03-0.3 ppb in October 1993 during POLINAT and Jurkat et al. (2010) measured 0.09 ppb of SO$_2$ during CONCERT in autumn 2008. Due to implementation of SO$_2$ emission control policies, the global SO$_2$ emissions decreased since 1980 (Hoesly et al., 2018; Aas et al., 2019), nevertheless the sulfur content in kerosene remained unchanged (Lee et al., 2021; Miller et al., 2009), thus the aviation based SO$_2$ emissions depend on the air traffic.

In 2020, a 72% reduction of the air traffic above Europe has been reported due to the COVID-19 lockdown in comparison to the same time period in 2019 (Schumann et al., 2021a, 2021b), hence providing a lower aviation SO$_2$ source with respect to 2019. Compared to 2010, the 3.5% increase in air traffic per year (Lee et al., 2011) promotes an increase by a factor of 1.5 of the 2010 air traffic for a scenario without COVID-19 restrictions in 2020 and consequently a theoretical increase of 50 percent in aviation SO$_2$ emissions for 2020, given that the sulfur content of the kerosene is unchanged (Lee et al., 2021; Miller et al., 2009). In 2020, The remaining air traffic of 28% (in comparison to 2019) corresponds to roughly 40% of the 2010 air traffic and might hence in part explain the SO$_2$ mixing rations detected in the upper troposphere, which are still higher than remote background measurements (Williamson et al., 2021).

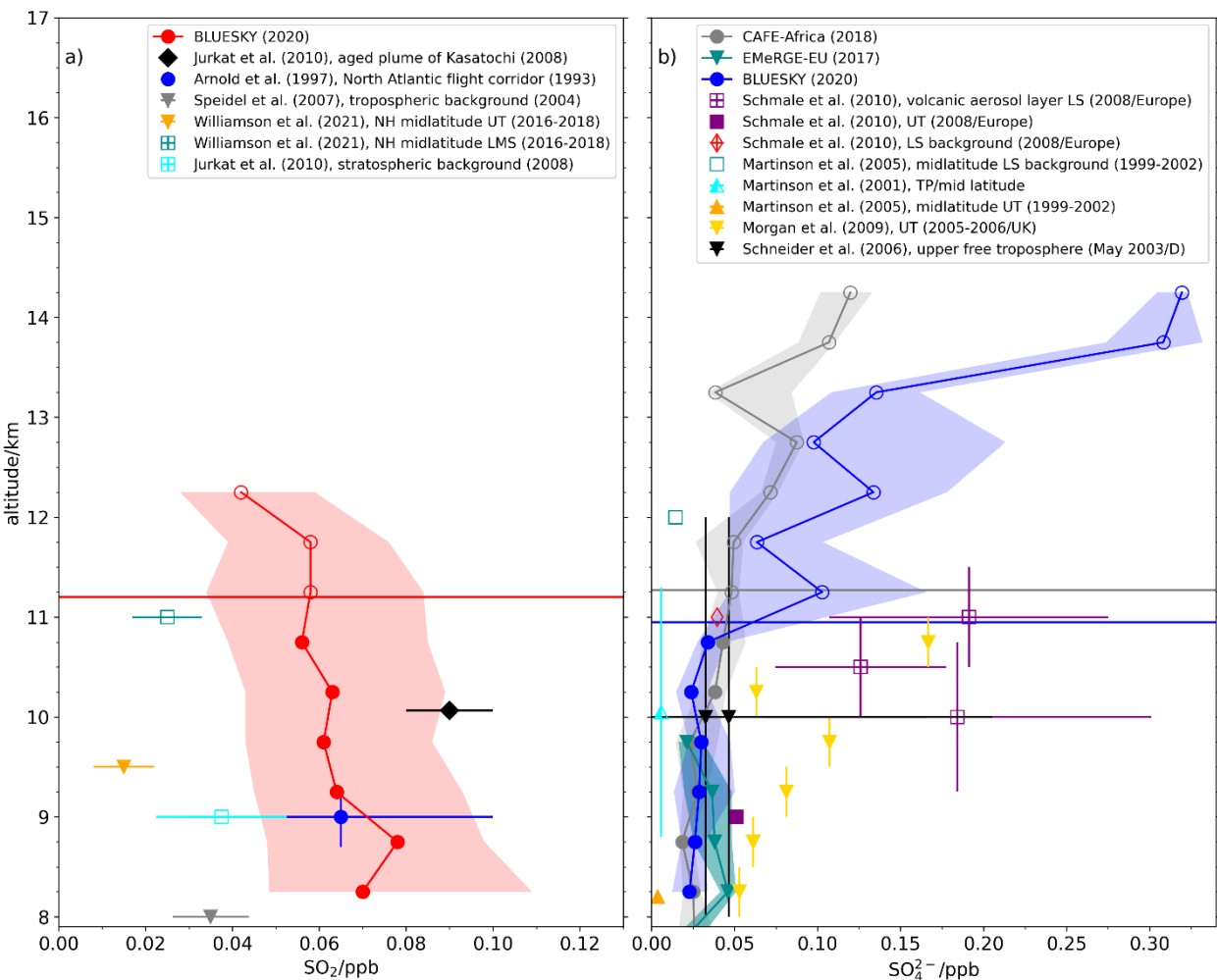

Figure 6: Profiles for a) $SO_2$ and b) $SO_4^{2-}$ for 500m altitude bins. In a) additional literature $SO_2$ values are shown, while in b) literature values for $SO_4^{2-}$ are added, including $SO_4^{2-}$ profiles from previous HALO missions with the aerosol mass spectrometer C-ToF-AMS onboard: EMeRGe-EU in June/July 2017 (Andrés Hernández et al., 2022) and four flights over Europe of the CAFE-Africa mission in July-September 2018. Full markers are tropospheric origin and open markers are stratospheric origin. Additionally, the dynamical tropopause (~2PVU) along the Falcon tracks is marked as red horizontal line in a) and the thermal tropopause is marked as blue horizontal line for HALO tracks and in grey for CAFE-Africa in b).

In addition, further sources could have contributed to the $SO_2$ budget in the upper troposphere. To analyse the origin of air masses with elevated $SO_2$, HYSPLIT back trajectories are calculated and representative examples are plotted in Fig. 5a-h. $SO_2$ emissions from anthropogenic and natural sources in the PBL can be lifted to the UT via convection or via warm conveyor belts and transported to the measurement region. Arnold et al. (1997) reported an extended layer of enhanced $SO_2$

with maxima of up to 3 ppb in the Northeast Atlantic, which was an air mass uplifted and transported from the polluted PBL from the eastern United States. A few cases show trajectories with similar pathways, like the example in Fig. 5e/f. Nevertheless, the PBL contacts are also over the Pacific and East Asia. The latter one suits better to the findings of Fiedler et al. (2009) who observed the uplift of polluted air masses from East Asia via warm conveyor belts and upper tropospheric

long-range transport towards Europe. Further, the Asian monsoon also serves as a vertical transport pathway for emissions from the PBL up to high altitudes, where the air mass can enter the LS and horizontally be transported either eastwards (Vogel et al., 2014, 2016) or can be horizontally transported in the UT (Tomsche et al., 2019) and finally reach Europe. Similar trajectory pathways can be found for the cases in Fig. 5a/b and 5g/h. These trajectories indicate long range transport in the UT and could have been impacted by the Asian monsoon. Generally, the trajectories with elevated $SO_2$ (Fig. 5a-h)

show lower potential temperatures in comparison to the trajectories calculated for elevated $SO_4^{2-}$ (Fig. 5i-p). Hence, long range transport of $SO_2$ enriched PBL air masses could have contributed to the observed BLUESKY $SO_2$ mixing ratios in the UT. In contrast, the trajectories do not indicate local transport from the central European PBL to the UT, hence the transport of $SO_2$ from wildfires in Germany in May 2020 (European Commission, 2021) to the UT seems negligible. Even if the transport of the smoke was via self-lofting (Ohneiser et al., 2021), i.e. absorption of sunlight leads to warming of the ambient

air and thus lifting of the smoke, the transport is slow and so $SO_2$ might already been transformed to $SO_4^{2-}$ before reaching the UTLS and does not contribute to the elevated $SO_2$ in the UT. Moreover, the decrease of $SO_2$ in the LS, as expected, does not support transport of $SO_2$ beyond the UT into the LS neither via convection nor warm conveyor belts. This can be confirmed by the trajectories for $SO_4^{2-}$ as they indicate long range transport at high altitudes with negligible influence from lower altitudes. The trajectories helped identifying potential source regions of $SO_2$.

Beside the sources, also sinks of $SO_2$ can alter the $SO_2$ concentrations in the UTLS. Beside the conversion to $H_2SO_4$, leading to sulfate particles, $SO_2$ is removed from the atmosphere by wet and dry deposition. $SO_2$ can be scavenged by clouds, which lead to a significant reduction of the $SO_2$ lifetime (Lelieveld, 1993). Van Heerwaarden et al. (2021) investigated the meteorological situation in spring 2020 and found that a stable high pressure system over Europe lead to a lower cloud fraction in comparison to the mean 2010-2019 period over Europe. This would lead to less cloud processing and reduce $SO_2$

sinks. Furthermore, elevated humidity favours the faster conversion of $SO_2$ to $SO_3$ and sulfate, as water vapour enhances the potential for elevated OH concentrations (Pandis and Seinfeld, 2006). As reported by Schumann et al. (2021a, 2021b) the UTLS was drier in spring 2020 in Europe in comparison to previous years. The median $H_2O$ profiles during BLUESKY reach only mixing ratios of up to 268 ppm in the upper troposphere (Fig. 3g), which is still within the range of typical springtime $H_2O$ mixing ratios in the upper troposphere (e.g. Hegglin et al., 2009; Kaufmann et al., 2018). But van

Heerwaarden et al. (2021) found with respect to humidity and cloud cover spring 2020 was amongst the springs with the lowest values. Thus, the lower available $H_2O$ led to lower OH concentrations during BLUESKY period, which implies less chemical processing and hence a reduction of $SO_2$ sinks. Less $SO_2$ sinks could lead to an enhanced $SO_2$ lifetime in the UTLS and thus higher $SO_2$ mixing ratios.

In sum, the enhanced $SO_2$ mixing ratios at cruise levels in Europe in spring 2020 can possibly be explained by a non-negligible aviation $SO_2$ contribution, WCB or convective transport from the boundary layer, followed by long range transport, and the prolonged $SO_2$ lifetime caused by the unusually dry UTLS conditions. Neither the sources nor the sinks could separately explain the $SO_2$ mixing ratios in the UTLS. Beyond that, we are not able to analyse in more detail the different amounts of the aforementioned factors and how they contribute to single flights.

## 5 Stratospheric sulfate aerosol

As mentioned in section 3.2, $SO_4^{2-}$ has a distinct profile with a steep increase at a potential temperature of 340 K, which refers here to around 11 km with respect to altitude (Fig. 6b). This altitude reflects also the thermal tropopause height during BLUESKY (Fig. 6b). Up to this altitude, the mixing ratio is rather constant, then it increases. Between 8-11 km $O_3$ mixing ratios are stable, and above $O_3$ increases. $O_3$ mixing ratio above 120 ppb indicates stratospheric air masses as mentioned above, thus the higher $SO_4^{2-}$ mixing ratios above 11 km can be attributed to the stratosphere and hence associated with stratospheric aerosol. The layer between 11 and 13.5 km can be influenced from the stratosphere as well as the troposphere, as the data are averaged over a few weeks and varying meteorological conditions, which lead to a broadening of the $25^{th}$ to $75^{th}$ percentiles range. In Figure 4a, this layer represents the mixed layer. The $SO_4^{2-}$ correlates well with $O_3$ for all flights, similar to the flight on 02 June 2020, presented in section 3.1 (Fig. 2).

Previous studies investigated the sulfate aerosol in the UTLS region in northern hemispheric midlatitudes (Fig. 6b). The BLUESKY mixing ratios in the UT agree well with the observations by Schneider et al. (2006) during May 2003, which were partly influenced by aircraft exhaust plumes. The BLUESKY mixing ratios are lower than the UT background reported by Schmale et al. (2010). Martinson et al. (2001, 2005) observed significantly lower $SO_4^{2-}$ mixing ratios in the European upper troposphere and tropopause region. The $SO_4^{2-}$ profile (Morgan et al., 2009) obtained from April 2005 to September 2006 over the UK shows higher $SO_4^{2-}$ concentrations compared to the BLUESKY measurements. Morgan et al. (2009) suggest that the elevated mixing ratios in the UT are the result of regional uplift of polluted air masses during stagnant meteorological conditions over the UK.

Sulfate was measured in two previous HALO missions with the aerosol mass spectrometer C-ToF-AMS in a similar altitude range and region. During EMeRGe-EU, seven research flights were conducted in June/July 2017 above Europe at altitudes up to 10 km (Andrés Hernández et al., 2022). The $SO_4^{2-}$ mixing ratio was on average $(0.04 \pm 0.01)$ ppb and compares well to the BLUESKY $SO_4^{2-}$ mean in the same altitude range below 10 km. The second HALO mission was CAFE-Africa in summer 2018 which reached altitudes up to 14 km. Here, only data obtained over Europe ($38°$ - $57°$N and $14°$W -$16°$E) are used for the comparison, which include two test flights and the ferry flights (27 July, 01 Aug, 07 Aug, and 07 Sept 2018; Voigt et al., 2022). For the CAFE-Africa subset the thermal tropopause was slightly higher than during BLUESKY (Fig. 6b). For the altitude range 8-11 km, the $SO_4^{2-}$ mean was $(0.03 \pm 0.01)$ ppb, similar to the BLUESKY value. Above 11 km in the

lower stratosphere, $SO_4^{2-}$ raises to $(0.09 \pm 0.03)$ ppb. Considering heights above the tropopause, i.e. with enhanced $SO_4^{2-}$ mixing ratios, the stratospheric BLUESKY $SO_4^{2-}$ concentrations are a factor of 2 to 3 higher than the observations in summer 2018.

In the following, we investigate the origin of the elevated stratospheric $SO_4^{2-}$ mixing ratios during BLUESKY. As mentioned above, we calculated HYSPLIT back trajectories for cases of elevated $SO_4^{2-}$. In Figure 5i-p, representative examples of

trajectories are displayed. The majority indicates long range transport at high altitudes with potential temperatures between 343 K and 465 K. While the lower range of the potential temperature is associated with midlatitude tropopause height, the upper values are clearly associated with the stratosphere. As only few trajectories indicate lower potential temperatures, we assume that the majority of elevated $SO_4^{2-}$ is already in the stratosphere 360 h before sampling and hence influence from the troposphere is negligible. One major source of $SO_4^{2-}$ in the stratosphere is volcanic eruptions. One year before BLUESKY,

the volcano Raikoke on the Kuril Islands (Russia, 48.29°N, 153.25°E) in the Western Pacific started to erupt on 21 June 2019 and continued for some days, it was categorised to volcanic explosivity index VEI≥4. Several explosive eruptions emitted a dense ash and $SO_2$ plume, which rose up to 19 km and 20 km on consecutive days (Hedelt et al., 2019), thus also injecting into the stratosphere. Based on TROPOMI analysis, de Leeuw et al. (2021) reported that the eruption released 1.4-1.6 Tg $SO_2$ into the atmosphere and simulated also that approximately 0.9-1.1 Tg $SO_2$ thereof were injected into the

stratosphere. Kloss et al. (2021) used satellite based OMPS (Ozone Mapping Profiler Suite Limb Profiler) stratospheric Aerosol Optical Depth (sAOD) to investigate the temporal evolution from before the Raikoke eruption until May 2020. Almost one year later, the sAOD was still higher than prior to the eruption. This suggests that elevated $SO_4^{2-}$ measured in the stratosphere during BLUESKY was partly caused by the Raikoke eruption a year earlier. The eruption of Mount St Helens in 1980 was of comparable size, midlatitude location, and $SO_2$ emissions (Deshler et al., 2006) and its impact was also visible

for almost a year. Still, other sources cannot be completely ruled out. For example, severe wildfires in Alberta/Canada developed pyro cumulus clouds in June 2019. The biomass burning emissions were uplifted into the lower stratosphere (Osborne et al., 2022). In July 2019, also severe fires in Siberia/Russia impacted the OPMS sAOD (Kloss et al., 2021). Ohneiser et al. (2021) discussed self-lofting as a potential transport pathway in the UTLS for these Siberian fires in the absence of strong vertical motion in July 2019. The smoke plume could raise and reach the UT and further ascent into the

LS. During the slow ascent, the emissions alter chemically, in the case of $SO_2$, it is transformed to $SO_4^{2-}$. Finally, the $SO_4^{2-}$ could have contributed to the enhanced $SO_4^{2-}$ in the LS. Further, wildfires in central Europe in May 2020 (European Commission, 2021) could also have undergone this self-lofting process as the trajectories do not indicate uplift over Europe and thus might additionally have contributed to elevated $SO_4^{2-}$ in the UTLS. In comparison to the Raikoke eruption, these biomass burning contributions are of lower magnitude. Reifenberg et al. (2022) suggest that other small and medium sized

volcanic eruptions from tropical latitudes, could have reached the stratosphere and thus impacted the stratospheric aerosol over Europe. One example is the volcano Taal on the Philippines (14.00°N, 120.99°E), which erupted on 12 January 2020, and its ash and gas plume rose up to around 10-15 km height (Global Volcanism Program, 2020. Report on Taal

(Philippines) ). The VEI was estimated to 3 and the $SO_2$ emissions estimated to 0.019 Tg (Liu et al., 2020). According to simulations of Reifenberg et al. (2022), the Taal eruption lead partly to an increase of $SO_4^{2-}$ in the LS during BLUESKY.

The measured $SO_4^{2-}$ mixing ratios in the LS agree with other volcano related in situ studies. The highest mixing ratios are reported by Schmale et al. (2010). They probed layers with enhanced $SO_4^{2-}$ in October/November 2008 roughly 3 months after the eruption of Mount Kasatochi (erupted 08 August 2008, 52.18°N, 175.51°W, 1.5 Tg $SO_2$), with an injection height reaching into the stratosphere and additionally Mount Okmok (53.40°N, 168.17°W) erupted on 12 July 2008 (0.2 Tg $SO_2$, Carn et al., 2008). Jurkat et al. (2010) also measured enhanced $SO_2$ concentrations in the stratosphere in the 3 months-old

Kasatochi plume during the CONCERT campaign (Voigt et al., 2010). Martinsson et al. (2009) reported particulate sulfur concentrations shortly after the Kasatochi eruption were 10 times higher than before the eruption and even 3-4 months after the eruption they were enhanced by a factor of 3. In contrast, during volcanic quiescent periods, like the period between 1997 and 2008 (Deshler, 2008) the $SO_4^{2-}$ has reduced mixing ratios in the stratosphere, and Martinson et al. (2005) reported $SO_4^{2-}$ mixing ratios of 0.01 ppb for the lower stratosphere for the years 1999-2002.

The enhanced $SO_2$ in the UT as described in section 4 and the longer $SO_2$ lifetime could possibly have a minor effect on the stratospheric sulfate aerosol. In these conditions, the $SO_2$ had more time to be transported into the LS and finally be transformed to $SO_4^{2-}$, adding to the $SO_4^{2-}$ mixing ratios. But it seems unlikely, as the trajectories for elevated $SO_4^{2-}$ (Fig. 5i-p) stay at high altitudes (and high potential temperatures) and thus mixing from the troposphere into the stratosphere is negligible. A few trajectories indicate a downward transport from higher altitudes, thus an origin deeper in the stratosphere

(Fig. 5k/l), which could be a hint for OCS as $SO_4^{2-}$ precursor. OCS is transported within the Brewer Dobson Circulation from the upper stratosphere to the lower stratosphere and being transformed via $SO_2$ to $H_2SO_4$. According to Brühl et al. (2012), it is the most important source for maintaining the stratospheric aerosol layer in volcanic quiescent periods and also for BLUESKY, OCS oxidation adds to the stratospheric $SO_4^{2-}$ background to some extent. Further, the Junge layer might also influence the $SO_4^{2-}$ mixing ratios in the LS. Even though the Junge layer is most pronounced at higher altitudes (Junge et al.,

1961), it could extend further down or due to the downward transport mentioned above (Fig. 5k/l) $SO_4^{2-}$ could be transported downward from the Junge layer into our measurement altitudes and thus contribute to the elevated $SO_4^{2-}$.

**6 Conclusion and outlook**

We find elevated $SO_4^{2-}$ mixing ratios in stratospheric air masses, and enhanced $SO_2$ mixing ratios in tropospheric air masses over Central Europe and the North Atlantic in spring 2020. The elevated $SO_2$ of 0.06 ppb in the UT agrees with $SO_2$ mixing

ratios performed in the background of the North Atlantic flight corridor in 2008 (Jurkat et al., 2010) despite lower air traffic due to COVID-19 restrictions in 2020. The 3.5% increase in air traffic since 2010 in part compensates the air traffic reduction in 2020. In addition, exceptional dry weather conditions leading to a low cloud fraction and low OH concentrations in the UTLS in May 2020 (Schumann et al., 2021a, 2021b, Van Heerwaarden et al., 2021) reduced $SO_2$ sinks and increased $SO_2$

lifetime. Back trajectories provided indications of other boundary layer $SO_2$ sources from convective or WCB transport and further long-range transport in the UT, which could have contributed to a small extend.

In the LS, enhanced $SO_4^{2-}$ mixing ratios were observed. In comparison to previous studies the $SO_4^{2-}$ mixing ratios were clearly above $SO_4^{2-}$ mixing ratios reported during volcanic quiescent periods (e.g. Martinson et al., 2001, 2005) and agreed with $SO_4^{2-}$ mixing ratios in volcanic impacted air masses (e.g. Schmale et al., 2010) measured at altitudes below 12 km. Compared to 2018, stratospheric $SO_4^{2-}$ was significantly enhanced in 2020. The eruption of the volcano Raikoke injected 0.9-1.1 Tg of $SO_2$ into the stratosphere (de Leeuw et al., 2021) in June 2019. In May 2020, an enhanced sAOD still was observed by Kloss et al. (2021) in the northern hemisphere caused by the Raikoke eruption and to a smaller extend by severe biomass burning events from June/July 2019 in Siberia and Canada. Further, Reifenberg et al. (2022) found that the eruption of the tropical volcano Taal in January 2020 contributed to the enhanced $SO_4^{2-}$ in the LS. We suggest these to be the primary sources of the enhanced stratospheric $SO_4^{2-}$ concentrations measured during BLUESKY, because back trajectories mainly showed long range transport in the lower stratosphere.

Overall, the unprecedented BLUESKY mission was conducted during exceptional meteorological conditions and also reduced air traffic, both impacted the $SO_2$ mixing ratios in the UT due to changes in the emissions and also sinks. The enhanced stratospheric sulfate aerosol, which was observed, was likely impacted by the volcano Raikoke, and smaller sources.

Together with the observations of other sulfur compounds such as gaseous $H_2SO_4$ on HALO, which are still under evaluation, the unique and comprehensive data set of sulfur compounds allows to test our understanding of the sulfur chemistry in global models (Reifenberg et al., 2022).

In a broader context, the present results give new insights in the sulfur chemistry in the UTLS region with respect to limited sources and sinks. They help to better understand a) the sensitivity of $SO_2$ to missing sinks and b) the stratospheric aerosol and its dependence on perturbations and their lasting impacts. Both aspects are important to improve models, especially with respect to simulations of the Earth's radiation budget, because changes in the radiation balance in the UTLS impact feedback processes in the global climate.

**Data availability**

Data are available on request at the HALO data base at https://halo-db.pa.op.dlr.de/mission/119.

**Author contribution**

CV, TJW, HS, JL planned the flight experiment. AM, JL, KK, JS, MS, LR performed the in-flight measurements. KK and JS provided evaluated particulate data and previous campaign data and supported the analysis. AM provided supporting evaluation and assisted the analysis. FO, AZ, HF, LR provided supporting evaluation. SK supported the analysis. LT evaluated

and analysed the data and prepared the manuscript with contributions from CV, AM, TJW, JS, KK, HF and SK. All co-authors
commented on the manuscript.

**Competing interests**

The authors declare that they have no conflict of interest.

**Acknowledgements**

Laura Tomsche is funded by the Deutsche Forschungsgemeinschaft (DFG, German Research Foundation) – TRR 301 –
Project-ID 428312742. The HALO flights during the BLUESKY mission were funded by the Max Planck Society. The authors
gratefully acknowledge the NOAA Air Resources Laboratory (ARL) for the provision of the HYSPLIT transport and
dispersion model and/or READY website (https://www.ready.noaa.gov) used in this publication. The authors thank Andreas
Giez, Christian Mallaun, and Vladyslav Nenakhov for providing the water vapor measurement onboard HALO and Falcon, as
well as Benedikt Steil for providing the ECMWF/ERA5 reanalysis data.

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
