# Peer review of "Enhanced sulfur in the UTLS in spring 2020"

_Atmospheric Chemistry and Physics, 2022_

## Author Comment (AC1)

**Comment on acp-2022-274**

Anonymous Referee #1
Referee comment on "Enhanced sulfur in the UTLS in spring 2020" by Laura Tomsche et al., Atmos. Chem. Phys. Discuss., https://doi.org/10.5194/acp-2022-274-RC1, 2022

**1) comments from referee**

This paper by Tomsche et al. presents new SO2 and SO42- (along with other trace gas species) measurements obtained during the BLUESKY aircraft research campaign in May/June 2020, during the European Covid-lockdown which offered an unprecedent opportunity to measure air masses characterised by drastically decreased air pollution. Interestingly, the authors find strongly enhanced SO2 and SO42- concentrations in the upper troposphere and lower stratosphere, respectively, despite overall reduced air pollutant emissions. While these observations are of great value and offer a benchmark against which the chemistry in air quality and other chemistry transport models can be tested, I find that the conclusions provided in the abstract not to be supported by enough evidence within the paper. The explanations for the enhanced sulfur-species concentrations comprise a range of different possibilities, which the authors fail to meaningfully constrain by their evaluations. I therefore cannot recommend this paper for publication in ACP in its current form.

**2) author's response**

We thank the reviewer for the comments. We went back to the analysis and followed your comments to improve the manuscript and to provide more evidence on the unique $SO_2$ and $SO_4$ data that we measured during spring 2020. We calculated more specific back trajectories and added more information, e.g. $H_2O$ profile and additional tropopause information, to ensure a more in-depth understanding of the whole meteorological situation. We hope that the revised version of the manuscript now more clearly supports the conclusions provided in the abstract In detail, we extended the explanations on the potential sinks and sources for $SO_2$ and set the chemical tropopause in the context of the thermal and dynamical tropopause. Moreover, we discussed air mass origins for elevated $SO_2$ and also $SO_4$ based on the extended trajectory analysis. Finally, we rephrased some of our argument to be easier readable and more comprehensible.

**Specific comments:**

**1) comments from referee**

L72, Section 2.1 BLUESKY mission:
It would be interesting for the reader to have a short summary of the mission goals added here. It seems obvious but it should be made more explicit as you have done nicely in the abstract.

**1a) author's response**

To give the reader a short overview of the goals of the missions, we added a short summary of the goals of the BLUESKY mission in Section 2.1

**1b) manuscript changes**

L78-81: "The period covered the first weeks of the COVID-19 lockdown in Europe and thus offered a unique opportunity to investigate an unprecedented situation of reduced anthropogenic emissions from industry, ground, and airborne transportation. The goal of BLUESKY was to explore the changes in the atmospheric composition and gain new insights on how anthropogenic emissions perturb chemical and physical processes in the atmosphere."

2) **comments from referee**

Figure 1. Could the paths of the different flights be represented in different linestyles or shades of blue/red to help emphasize that they were carried out on different days? The current figure may show the coverage, is otherwise though not very informative.
Also, you talk about coordination of the flights between Falcon and HALO, but the coverage is rather different. What was the main aim of the coordination?

2a) **author's response**

We agree with the reviewer that the reader might get the impression that HALO and Falcon flight tracks differ from each other. Despite different aircraft ranges of HALO and Falcon (max altitude of 14.5 km and 12.5 km and around 10 and 4 flight hours, respectively), we succeeded to have 5 out of 8 HALO missions together with HALO and Falcon. Here we focus on 20 flights that were flown with both aircraft over the course of BLUESKY, while the majority of flights were combined. This was made possible due to combined flight planning for HALO and Falcon. During these flights both aircraft probed similar air masses or the same region. In order to account for differences in endurance and flight velocity, We took profit from the different ranges and in some occasion HALO flew above Falcon to extend the range and Falcon performed profiling measurements during refuelling. To make this more clear, we updated Figure 1 and adapted the caption and added text in section 2.1

2b) **manuscript changes**

L81-83: "The payload of both aircraft was complementary to obtain a comprehensive dataset. Especially, five days with coordinated flights over Germany (23 May, 26 may, and 28 May) and over the North Atlantic (30 May, 02 June), the payload offered the opportunity to probe the air masses in more detail."

**Caption L91-92: "…Coordinated flights (Falcon: orange; HALO: cyan) were performed over Germany on 23 May, 26 May, and 28 May and twice (30 May and 02 June) as both aircraft headed towards the North Atlantic, west of Ireland. Coordinated flights (Falcon: orange; HALO: cyan) were performed over Germany on 23 May, 26 May, and 28 May and twice (30 May and 02nd June) as both aircraft headed towards the North Atlantic, west of Ireland. "**

3) **comments from referee**

L96 Sensitivity of measurement to moisture: It seems somewhat arbitrary to use a specific altitude as cut-off since you could find samples with high/low moisture content even below/above 8 km depending on the meteorological situation you're flying in. What is the range of H2O mixing ratios you can/cannot easily perform this correction for? Did you measure H2O and if yes, with which instruments?

3a) **author's response**

On board Falcon, $H_2O$ was measured with the onboard Lyman-alpha absorption instrument, which is part of the meteorological sensor system.

For the BLUESKY $SO_2$ data, the moisture correction could be applied for water vapor mixing ratios roughly up to 500 ppm, which corresponds to an altitude of ca. 7.5 km. With higher water vapor mixing ratios, the uncertainty in the analysis increases. The uncertainty of the correction is around 1.4%, for the low water vapor mixing ratios, for higher water vapor the uncertainty dominates and leads to an increase of the total uncertainty. The uncertainty of the correction is quantified in the uncertainty analysis. Additionally, fast ascends and descends impact the data quality and thus the uncertainty.

To limit the impact of both effects, we restricted our analysis to above 8 km as the focus on the presented study is the UTLS region. We adapted the text

3b) **manuscript changes**

L105-106: "…, a correction is more difficult and reduces the data quality. As the focus of the present study is the UTLS region, we limited our analysis on altitudes above 8 km and thus ensure the data quality. "

L111: "…The total uncertainty is on average 22.7% for SO2 and included the uncertainty of the moisture correction."

L115-116: "Additionally, water vapor ($H_2O$) was measured with the Lyman-alpha absorption instrument integrated in the meteorological sensor system."

4) **comments from referee**

L138 Could be written more clearly. I suggest to replace 'along the same longitudes and vice versa' with 'and vice versa along the longitudes outside this range'

4a) **author's response**

I implemented your suggestion.

4b) **manuscript changes**

L149: "… and 28 ppb (HALO) and vice versa along the longitudes outside this range, when CO mixing ratios were enhanced…"

5) **comments from referee**

L142 Related comment. I suggest to explicitly say that O3 and HNO3 are positively correlated as expected and repeat the longitude range here, since 'in the mentioned longitude range' may not be clear to readers given that you talk about two in L138.

5a) **author's response**

Following the suggestion, we changed the text

5b) **manuscript changes**

L152-153 to:" Between 6°W and 3°E, $O_3$ and $HNO_3$ are positively correlated, as expected, and $HNO_3$ mixing ratios increase up to 1.6 ppb."

6) **comments from referee**

L148 I would rewrite this sentence here to point towards the more in-depth analysis and discussion in Section 5 and without claiming it is 'just' from long-range transport. As it currently stands here, I cannot judge from Figure 5 whether the evidence you provide is good enough to underpin this result. For example, I see one trajectory rising from rather low altitude starting around the Eastern coast of North America ending at the measurement location.

6a) **author's response**

We agree, that mentioning long range transport in this section is not the right spot and an indication for further explanations later on fits better. Now the last sentence reads:

6b) **manuscript changes**

L159: "An in-depth analysis and discussion on potential explanations for this or similar features will follow in section 4."

7) **comments from referee**

Figure 3: I suggest adding the tropopause height onto the figure.

7a) **author's response**

The dynamical tropopause as 2PVU is now added to Figure 3 including updated caption.

7b) **manuscript changes**

L199-200 (caption): "Additionally, the black line at 335 K roughly indicates the dynamical tropopause (as 2 PVU) based on ECMWF/ERA5 analysis along the Falcon flight tracks."

8) **comments from referee**

L163 It would seem important to indicate the average altitude of the dynamical and/or thermal tropopauses over the region during this time period as well, not to give the impression of choosing what fits best your lower bound of the mixing layer. I would expect the 2PVU tropopause being close to the 330K isentrope, so it would confirm your choice of ozone value for defining the chemical tropopause.

8a) **author's response**

The dynamical tropopause is based on ECMWF/ERA5 analysis along the Falcon flight tracks. The median theta profile exceeds at 335K the 2 PVU. After adding in Figure 3 the dynamical tropopause height, which confirmed our choice of the chemical tropopause at 120 ppb $O_3$, we added a sentence in the text.

8b) **manuscript changes**

L174-178: "The chemical tropopause is marked by strong gradients in several tracers. O3 as a common indicator shows a kink in its median profile at around 140 ppb and 340 K potential temperature, which is within the limits given by Thouret et al. (2006). The dynamical tropopause, displayed as the 2 PVU based on ECMWF/ERA5 data along the Falcon flight tracks (Fig. 3), is around 335 K and thus has a similar height as the chemical tropopause."

9) **comments from referee**

L203 Is this an expected concentration range for the mixing layer to be found in at these latitude bands and season?

9a) **author's response**

Yes, it is within the range of latitude and season. Pan et al. (2004) and Hoor et al. (2002) showed a similar extension of the transition layer at midlatitudes in summer. I added some references in the text.

9b) **manuscript changes**

L217-219: "The mixing layer almost extends over the whole $O_3$ range from 150 ppb to 400 ppb, similar to other mixing layers in the same latitude and season (Hoor et al., 2002; Pan et al., 2004) and thicker in comparison to a winter polar mixing layer (Fischer et al., 2000)."

10) **comments from referee**

Figure 4 and discussion section 3.3: It is really hard to follow the discussion of this figure. The only really outstanding feature I can detect when looking at these panels is that there are some high concentrations in SO42- in the troposphere (at ozone values below 100 ppbv and CO values between 100 and 125 ppbv). It would be nice to have this feature explained by the backward trajectories. Otherwise, it is expected that SO2 decreases and SO42- increases as one goes into the stratosphere due to the aging of air, which is reflected both in the very strong anticorrelation between the two sulfur species and also in the SO42- / (SO42- + SO2) ratio visible in Figure 2 and 3. Maybe you could circle the points you are referring to in the figure in case I have missed what you are really referring to?

10a) **author's response**

Thanks for pointing to the feature. The outstanding feature in the $SO_4^{2-}$, you mentioned, is out of the scope of this study and should not be included in the data. In the updated Figure, it is removed to not confuse the reader. Additionally, the outlier mentioned in L224-225 is now circled in Fig. 4c. Figure 4 is updated including the caption.

10b) **manuscript changes**

**L243-246 (caption): "Figure 4: Tracer-tracer correlation for a) 30 sec data CO - $O_3$ in black for whole altitude range and for heights above 8 km $SO_4^{2-}$ is color-coded from HALO flights, b) CO - $O_3$ in black for whole altitude range, and with color-coded $SO_2$ (above 8 km) from Falcon flights when $O_3$ was available, and c) 1 sec data CO - $HNO_3$ with color-coded $SO_2$ from Falcon flights. In c) a grey circle marks an outlier with high $SO_2$, CO and $HNO_3$."**

L229-230: "One $SO_2$ outlier with higher mixing ratios at enhanced $HNO_3$ und reduced CO can be identified (Fig 4c; grey circle)."

L239-240: "The back trajectories of $SO_2$ and $SO_4^{2-}$ cases support the assumption that the origins differ, especially concerning the different altitude ranges of the trajectories (Fig. 5)."

11) **comments from referee**

L215/Figure 5. This evaluation would be more effective and convincing if you would identify the exact times/altitude/geographical location of when unusual trace gas observations are made and then calculate your backward trajectories from locations on a finer grid point around these. I then would suggest not only to show lat-lon plots, but also time-altitude evolution of the trajectories and also PV along with it, so that the reader can see where/when tropospheric influence or strong uplift within the Asian monsoon may have happened.

11a) **author's response**

In the new Figure 5, exact time/altitudes/geographical location and also $SO_2$ or $SO_4$ mixing ratios are included. It includes lat-lon, as well as hour before release-altitude plots. HYSPLIT has no tropopause height information, thus the closest indication is the potential temperature, which is given as color-coded tracks. Text and caption are adapted.

11b) **manuscript changes**

L232-234: "… long range transport in the UTLS region for this case (Fig. 5a-b). Generally, the trajectories do not indicate transport from local PBL sources for cases with elevated $SO_2$. In Figure 5ah, examples for cases with elevated $SO_2$ are shown, including the case on 02 June 2020 described in section 3.1 (Fig. 5g/h)."

**L250-255: "Figure 5: HYSPLIT 360 hours back trajectories calculated for cases with elevated $SO_2$ (a-h; Falcon) and $SO_4^{2-}$ (i-p; HALO) mixing ratios. The release points started in the vicinity of these events. In the left column: in black are the flight tracks, color-coded is the potential temperature along the trajectories to indicate the transport altitude. The right column represents the trajectories as hours before release vs. altitude, also with color-coded potential temperature The cases of enhanced $SO_2$ were on 19 May 2020 (a, b), 23 May 2020 (c, d), 30 May 2020 (e,f), and 02 June 2020 (g, h). The cases of elevated $SO_4^{2-}$ were on 23 May 2020 (i, j), 26 May 2020 (k, l), 30 May 2020 (m, n) and 02 June 2020 (o, p)."**

12) **comments from referee**

Figure 5 Why is there an empty panel (j)?

12a) **author's response**

Figure 5 is replaced by a new one including other trajectories.

13) **comments from referee**

L253-262 I cannot fully follow your argumentation here. Don't you miss to account for the 72% reduction in air traffic due to Covid when you calculate your 50% increase in aviation SO2 emission increases?

13a) **author's response**

I see your point and formulated the argumentation more precisely. The 50% increase is not related to the 72% reduction, but is a theoretical value for a scenario without COVID19 restrictions. I adapted the paragraph

13b) **manuscript changes**

L280-285: "…by a factor of 1.5 of the 2010 air traffic for a scenario without COVID19 restrictions in 2020 and consequently a theoretical increase of 50 percent in aviation $SO_2$ emissions for 2020, given that the sulfur content of the kerosene is unchanged (Lee et al., 2021; Miller et al., 2009). In 2020, The remaining air traffic of 28% (in comparison to 2019) corresponds to roughly 40% of the 2010 air traffic and might hence in part explain the $SO_2$ mixing rations detected in the upper troposphere, which are still higher than remote background measurements (Williamson et al., 2021)."

14) **comments from referee**

L291-295 This conclusion is not very satisfying. Much more should have been done to better disentangle all the different possible explanations of the enhanced SO2 throughout the upper troposphere. The reader knows in principle as much as before reading your paper about the potential sources of the enhanced sulfur species. If you could do the backward trajectories in a more targeted way, that is identifying the origin of high-SO2 events for different flights more meaningfully, this could be a much better step to answer the question you wish to answer.

14a) **author's response**

We added a $H_2O$ profile in Fig. 3g as suggested by the 2nd reviewer, to stress the explanation of the $SO_2$ sinks and added for the $H_2O$ measurement Martin Zöger as coauthor. Further, we calculated more specific trajectories for high $SO_2$ and also high $SO_4$ cases and replaced Fig. 5 with a new one, which

includes representative examples of back trajectories. We adapted the text with more in-depth analysis regarding trajectories in section 4, especially we rewrote the paragraphs about $SO_2$ sinks, $SO_2$ sources, and the conclusion including new trajectory analysis.

**14b) manuscript changes**

L137-139: "For selected cases with either elevated $SO_2$ or $SO_4^{2-}$ mixing ratios, 360 h back trajectory ensembles were calculated. One ensemble consists of 27 single trajectories, which are offset by one meteorological grid point in the horizontal and 0.01 sigma in the vertical coordinate."

L 295-313: "In addition, further sources could have contributed to the $SO_2$ budget in the upper troposphere. To analyse the origin of air masses with elevated $SO_2$, HYSPLIT back trajectories are calculated and representative examples are plotted in Fig. 5a-h. $SO_2$ emissions from anthropogenic and natural sources in the PBL can be lifted to the UT via convection or via warm conveyor belts and transported to the measurement region. Arnold et al. (1997) reported an extended layer of enhanced $SO_2$ with maxima of up to 3 ppb in the Northeast Atlantic, which was an air mass uplifted and transported from the polluted PBL from the eastern United States. A few cases show trajectories with similar pathways, like the example in Fig 5e/f. Nevertheless, the PBL contacts are also over the Pacific and East Asia. The latter one suits better to the findings of Fiedler et al. (2009) who observed the uplift of polluted air masses from East Asia via warm conveyor belts and upper tropospheric long-range transport towards Europe. Further, the Asian monsoon also serves as a vertical transport pathway for emissions from the PBL up to high altitudes, where the air mass can enter the LS and horizontally be transported either eastwards (Vogel et al., 2014, 2016) or can be horizontally transported in the UT (Tomsche et al., 2019) and finally reach Europe. Similar trajectory pathways can be found for the cases in Fig. 5a/b and 5g/h. These trajectories indicate long range transport in the UT and could have been impacted by the Asian monsoon. Generally, the trajectories with elevated $SO_2$ (Fig. 5a-h) show lower potential temperatures in comparison to the trajectories calculated for elevated $SO_4^{2-}$ (Fig. 5i-p). Hence, long range transport of $SO_2$ enriched PBL air masses could have contributed to the observed BLUESKY $SO_2$ mixing ratios in the UT. Nevertheless, the decrease of $SO_2$ in the LS, as expected, does not support transport of $SO_2$ beyond the UT into the LS neither via convection nor warm conveyor belts. This can be confirmed by the trajectories for $SO_4^{2-}$ as they indicate long range transport at high altitudes with negligible influence from lower altitudes. The trajectories helped identifying potential source regions of $SO_2$."

L314-327: "Beside the sources, also sinks of $SO_2$ can alter the $SO_2$ concentrations in the UTLS. Beside the conversion to $H_2SO_4$, leading to sulfate particles, $SO_2$ is removed from the atmosphere by wet and dry deposition. $SO_2$ can be scavenged by clouds, which lead to a significant reduction of the $SO_2$ lifetime (Lelieveld, 1993). Van Heerwaarden et al. (2021) investigated the meteorological situation in spring 2020 and found that a stable high pressure system over Europe lead to a lower cloud fraction in comparison to the mean 2010-2019 period over Europe. This would lead to less cloud processing and reduce $SO_2$ sinks. Furthermore, elevated humidity favours the faster conversion of $SO_2$ to $SO_3$ and sulfate, as water vapour enhances the potential for elevated OH concentrations (Pandis and Seinfeld, 2006). As reported by Schumann et al. (2021a, 2021b) the UTLS was drier in spring 2020 in Europe in comparison to previous years. The median $H_2O$ profiles during BLUESKY reach only mixing ratios of up to 268 ppb in the upper troposphere (Fig 3g), which is still within the range of typical springtime $H_2O$ mixing ratios in the upper troposphere (e.g. Hegglin et al., 2009; Kaufmann et al., 2018). But van Heerwaarden et al. (2021) found with respect to humidity and cloud cover, spring 2020 was amongst the springs with the lowest values. . Thus, the lower available $H_2O$ led to lower OH concentrations during BLUESKY period, which implies less chemical processing and hence a reduction of $SO_2$ sinks. Less $SO_2$ sinks could lead to an enhanced $SO_2$ lifetime in the UTLS and thus higher $SO_2$ mixing ratios. "

L328-332: "In sum, the enhanced $SO_2$ mixing ratios at cruise levels in Europe in spring 2020 can possibly be explained by a non-negligible aviation $SO_2$ contribution, WCB or convective transport

from the boundary layer, followed by long range transport, and the prolonged $SO_2$ lifetime caused by the unusually dry UTLS conditions. Neither the sources nor the sinks could separately explain the $SO_2$ mixing ratios in the UTLS. Beyond that, we are not able to analyse in more detail the different amounts of the aforementioned factors and how they contribute to single flights. "

**15) comments from referee**

Section 5: It would be interesting to see whether the clear anti-correlation in SO42- with ozone seen for the flight in Figure 2 is also found for other flights. This could then be used to identify truly anomalous correlations for which again more targeted backward trajectories could potentially yield much more insight than is provided here. As the discussion is presented here, multiple explanations from existing literature but no additional insights are provided, leaving the reader once more unsatisfied about having learned anything new.

**15a) author's response**

I suppose that the referee means the correlation between $O_3$ and $SO_4$. $SO_4$ is clearly correlated with $O_3$ for all flights in the UTLS region presented here, comparable to the flight presented in Figure 2. In order to provide further insight in the airmass origins of cases of elevated $SO_4$, we calculated specific trajectories. In Figure 5i-p four representative cases are displayed.

**15b) manuscript changes**

**L251-256 (caption): "Figure 5: HYSPLIT 360 hours back trajectories calculated for cases with elevated $SO_2$ (a-h; Falcon) and $SO_4^{2-}$ (i-p; HALO) mixing ratios. The release points started in the vicinity of these events. In the left column: in black are the flight tracks, color-coded is the potential temperature along the trajectories to indicate the transport altitude. The right column represents the trajectories as hours before release vs. altitude, also with color-coded potential temperature The cases of enhanced $SO_2$ were on 19 May 2020 (a, b), 23 May 2020 (c, d), 30 May 2020 (e,f), and 02 June 2020 (g, h). The cases of elevated $SO_4^{2-}$ were on 23 May 2020 (i, j), 26 May 2020 (k, l), 30 May 2020 (m, n) and 02 June 2020 (o, p)."**

L341-342: "In Figure 4a, this layer represents the mixed layer. The $SO_4^{2-}$ correlates well with $O_3$ for all flights, similar to the flight on 02 June 2020, presented in section 3.1 (Fig. 2)."

L362-368: "As mentioned above, we calculated HYSPLIT back trajectories for cases of elevated SO42-. In Figure 5i-p, representative examples of trajectories are displayed. The majority indicates long range transport at high altitudes with potential temperatures between 343 K and 465 K. While the lower range of the potential temperature is associated with midlatitude tropopause height, the upper values are clearly associated with the stratosphere. As only few trajectories indicate lower potential temperatures, we assume that the majority of elevated SO42- is already in the stratosphere 360 h before sampling and hence influence from the troposphere is negligible."

**16) comments from referee**

L360 I do not think that you have provided real evidence that you have measured enhanced SO42- concentrations in the stratosphere, since we do not know what the 'normal' background would be. The difference seen in Figure 6 could simply be due to measuring within the troposphere at altitudes 11-15 km for the CAFE-Africa campaign data, which to my understanding was performed within the tropics.

**16a) author's response**

Yes, CAFE-Africa was performed in the tropics, but we only used a subset of flights north of 38°N as described in L354 -356 to be able to compare to BLUESKY. For a better comparability of stratospheric air masses, I added the tropopause height for BLUESKY and CAFE-Africa in Fig. 6.

Further, we also compared the $SO_4$ mixing ratios with studies of volcanic quiescent periods and other available studies, to receive a bigger picture about the situation in spring 2020 in comparison to other situations (e.g. Martinson et al., 2005 in L396).

I adapted the text. Moreover, the new trajectory calculation, as mentioned above, provide more insight in the origin of the elevated $SO_4$. Altogether it improved our analysis and helped to provide more evidence of the measured $SO_4$.

Changes in the text are given in the previous comment/answer section concerning trajectories with respect to elevated $SO_4$ cases, and additional changes in the text are the following:

16b) **manuscript changes**

L335-336:". This altitude reflects also the thermal tropopause height during BLUESKY (Fig. 6b)."

L357: "For the CAFE-Africa subset the thermal tropopause was slightly higher than during BLUESKY (Fig. 6b)."

L401-404: ". But it seems unlikely, as the trajectories for elevated $SO_4$ (Fig 5i-p) stay at high altitudes and thus mixing from the troposphere into the stratosphere is negligible. A few trajectories indicate a downward transport form higher altitude, thus an origin deeper in the stratosphere (Fig 5k/l), which could be a hint for OCS as $SO_4^{2-}$ precursor."

L418-419: "Back trajectories provided indications of other boundary layer $SO_2$ sources from convective or WCB transport and further long-range transport in the UT, which could have contributed to a small extend."

And L427-429: "… primary sources of the enhanced stratospheric $SO_4^{2-}$ concentrations measured during BLUESKY, because back trajectories mainly showed long range transport in the lower stratosphere. "

*Minor corrections/typos:*
17) **comments from referee**

L31 geo engineering --> geoengineering

17a) **author's response**

 Done L32
18) **comments from referee**

L90 in flight --> in-flight

18a) **author's response**

 Done L98
19) **comments from referee**

L146 delete 'to' in front of '266 ppb…'

19a) **author's response**

 Done L157
20) **comments from referee**

L179 write either 'with stratospheric sulfate aerosol' or 'with the stratospheric sulfate aerosol layer'

20a) **author's response**

L194: changed to 'with stratospheric sulfate aerosol'

**Reviewer 2 (Daniele Visioni):**

21) **comments from referee**

Referee comment on "Enhanced sulfur in the UTLS in spring 2020" by Laura Tomsche et al., Atmos. Chem. Phys. Discuss., https://doi.org/10.5194/acp-2022-274-RC2, 2022

This manuscript is overall an excellent overview of the BLUESKY mission, and presents some rather interesting measurements of the UTLS in a peculiar case (the beginning of the COVID-driven decrease in emission). It has the potential to be a rather important reference going forward, but clearly the manuscript needs more work, and to be cleaned up a bit.

21a) **author's response**

Thanks for writing a review. I spend more time analyzing data, especially back trajectories to improve the manuscript and I rephrased some paragraphs to be more precisely in the explanations.

22) **comments from referee**
I generally agree with all the comments already laid out by Reviewer 1, so I find it pointless to repeat them one by one: but in particular, I found the possible causes of enhanced SOA,, to be a bit too much handwaved in section 4.

For instance, at L. 283, the paragraph starts with "Beside the conversion to H2SO4..." but then continues by mentioning that most of the SOA,, would be reduced by its conversion to sulfates through OH oxidation. Something you clearly go back to at the end of the paragraph, by simply repeating the same thing. But no convincing proofs are given. Even just the addition of a vertical profile of humidity from ERA5 reanalysis (which is what van Heerwaarden et al. uses) would go a long way into proving that, indeed, SOA, lifetime has been increased by lower OH concentrations.

22a) **author's response**

Thank you for the idea of a $H_2O$ profile. I decided to use in situ data onboard HALO and Falcon in Fig. 3g. OH was also measured during BLUESKY onboard HALO, but the data are not finalized yet and are not available for the present work. Nevertheless, the OH trend supports our interpretation. Added text and rewrote the paragraph about the $SO_2$ sinks.

22b) **manuscript changes**

L 115-116: "Additionally, water vapor ($H_2O$) was measured with the Lyman-alpha absorption instrument integrated in the meteorological sensor system."

L 130-131: "Water vapor was measured with the tunable diode laser (TDL) hygrometer SHARC (Sophisticated Hygrometer for Atmospheric ResearCh)."

L180-182 "The profiles of the tropospheric tracer $H_2O$ also decrease with height, following the CO profile, but with a less pronounced step around the chemical tropopause. The $H_2O$ mixing ratios are 26-268 ppb (Falcon) and 3-153 ppb (HALO)."

L314-327:" Beside the sources, also sinks of $SO_2$ can alter the $SO_2$ concentrations in the UTLS. Beside the conversion to $H_2SO_4$, leading to sulfate particles, $SO_2$ is removed from the atmosphere by wet and dry deposition. $SO_2$ can be scavenged by clouds, which lead to a significant reduction of the $SO_2$ lifetime (Lelieveld, 1993). Van Heerwaarden et al. (2021) investigated the meteorological situation in spring 2020 and found that a stable high pressure system over Europe lead to a lower cloud fraction in comparison to the mean 2010-2019 period over Europe. This would lead to less cloud processing and reduce $SO_2$ sinks. Furthermore, elevated humidity favours the faster conversion of $SO_2$ to $SO_3$ and sulfate, as water vapour enhances the potential for elevated OH concentrations (Pandis and Seinfeld, 2006). As reported by Schumann et al. (2021a, 2021b) the UTLS was drier in spring 2020 in Europe in comparison to previous years. The median $H_2O$ profiles during BLUESKY reach only mixing ratios of up to 268 ppb in the upper troposphere (Fig. 3g), which is still within the range of typical springtime $H_2O$ mixing ratios in the upper troposphere (e.g. Hegglin et al., 2009; Kaufmann et al., 2018). But van Heerwaarden et al. (2021) found that the humidity and the cloud cover of spring 2020 were amongst the springs with the lowest values. Thus, the available $H_2O$ lead to lower OH concentrations during BLUESKY period and would imply less chemical processing and hence would also reduce $SO_2$ sinks. Less $SO_2$ sinks could lead to an enhanced $SO_2$ lifetime in the UTLS and thus higher $SO_2$ mixing ratios. "

23) **comments from referee**
Something similar can be said for Section 5.

A rather long list is given, but I don't find a compelling critical assessment of possible causes. Starting from the bottom, sure, we know COS contributes to the stratospheric aerosol layer, but what causes would have produced an increase significant enough to be observed in this campaign (noting that photolysis of OCS happens much higher up, so its eventual conversion to aerosols and signal at the lower altitudes would not be as instantaneous as for SOA). So in all of this, Figure 6 is really not that helpful (also, the legend in panel b) seems to be hiding the 2017 data...). Are we sure those different measurements are performed over similar regions? The Andrés Hernández (2022) reference seems to be targeted at lower altitudes, and I can't find SO4 measurements there, so it's hard to judge if the comparison is correct or not - and it's the only point offered for comparison! This hand-waviness is reproduced as is in the Conclusions, where the phrase " The enhanced stratospheric sulfate aerosol, which was observed, was likely impacted by the volcano Raikoke, and smaller sources" doesn't really say anything at all.

23a) **author's response**

Andrés Hernández et al. (2022) presented the EMeRGe-EU campaign in summer 2017 with measurements over Central Europe, even though the measurements are limited to an altitude of 10 km, the data provide an additional comparison for the $SO_4$ profile in the upper troposphere. In the UT, their $SO_4$ mixing ratios agree with the BLUESKY data well. The European subset of the CAFE-Africa data set is another comparison (which has not been published before) and covers the same altitude range than the BLUESKY mission, thus offering the option to compare UT and LS $SO_4$ mixing ratios.

Figure 6 is updated and the legend is not hiding any data. Further, I included for the BLUESKY Falcon flights (a) and HALO flights (b) a dynamical tropopause (~2 PVU) and a thermal tropopause, respectively. In b) also the thermal tropopause for the CAFE-Africa subset is included. Further comparison to other studies is given in the text and Fig 6b) (e.g. Schmale et al., 2010). Moreover, an additional potential source, which could contribute to elevated SO4 is added in the text:

23b) **manuscript changes**

L407-410: "Further, the Junge layer might also influence the $SO_4^{2-}$ mixing ratios in the LS. Even though the Junge layer is most pronounced at higher altitudes (Junge et al., 1961), it could extend further down or due to the downward transport mentioned above (Fig. 5k/l) $SO_4^{2-}$ could be transported downward from the Junge layer into our measurement altitudes and thus contribute to the elevated $SO_4^{2-}$."

24) **comments from referee**
In a future revision, both Section 4 and 5 need to be strengthened and more analyses need to be performed to make sure this really reads like a Research Article. Otherwise, the editor and the authors could consider shortening those parts and limiting the scope of the work to a "Measurement report" type of manuscript for ACP.

24a) **author's response**

I followed your suggestion and performed more analysis, especially more trajectory analysis, to get a better understanding on the air mass origins of elevated $SO_2$ and also $SO_4$ cases. Further, I included $H_2O$ profiles to strengthen the assumption about the exceptional dry weather situation in the UTLS as mentioned in other studies. I think the manuscript improved to present a research article rather than a measurement report.

25) **comments from referee**

One more typo: "One major source of $SO_4^{2-}$ in the stratosphere are volcanic eruptions" should be "is" and not "are"

25a) **author's response**

Done L368

---

## Referee Report (RR1)

This manuscript provides an overview of the BLUESKY mission during the 2020 COVID-19 lockdown in Europe. The authors present interesting measurement results of $SO_2$ and $SO_4^{2-}$, along with other trace gases and aerosols, at UTLS, which have the potential to be an important reference to future chemistry and modeling development. The authors have revised the manuscript with more discussions based on the comments from previous reviewers.

In addition to the comments from previous reviewers, I'm particularly interested in how wildfires contribute to $SO_2/SO_4^{2-}$ profile changes. Although the authors cited previous studies that demonstrated smaller amount of $SO_2$ released from wildfires compared to that from volcanic eruptions, this wouldn't necessarily represent the BLUESKY case since the meteorological conditions may be different between 2019 and 2020. Based on the 2020 EU JRC wildfire report (https://ec.europa.eu/commission/presscorner/detail/en/ip_21_5627), there were still significant wildfire events in Europe, especially in Germany, during May 2020. If those wildfires were intense enough, $SO_2$ and $SO_4^{2-}$ at UTLS could be influenced. The authors should provide sufficient evidence to convince readers how these "local" wildfires, in addition to long-range transport $SO_2$ from other continents, didn't contribute significantly to the $SO_2$ and $SO_4^{2-}$ profile changes during the BLUESKY mission. Additional discussion about wildfires can help improve the manuscript.

**Minor comments:**

Ln 22, 124, 125, 128, 353, 358, 359. Keep the number expression consistent. A thin space is suggested before and after the plus-minus sign.

Ln 55. "($46°N$, 0.8 Tg $SO_2$)" is suggested.

Ln 59. Typo: "important source of stratospheric aerosol are intense wildfires, ..." should be "is" instead of "are."

Fig. 1 The bottom axis at 0 longitude seems to have an additional character embedded.

Ln 64. You may want to spell out COVID at its first appearance in the text. Also, be consistent with using COVID-19 or COVID19 in the manuscript.

Ln 128. Lower case of N for "nitrous."

Ln 132. This sentence should be combined with the previous sentence.

Ln 137. A proper citation/reference to the GDAS dataset is mandatory.

Ln 159. May replace "will follow" with "is presented."

Fig.2 caption. "… Plotted are a) $SO_2$, …, and f) altitude across longitudes ##°W–##°E."

Fig. 3 caption. Superscripts for $25^{th}$ and $75^{th}$, be consistent with how they are used in the main text.

Ln 205. "… 310 K and 340 K; above the chemical tropopause the sum increases up to …"

Ln 395. Typo: "were" instead of "where."

---

## Author Response (AR3)

Comment on acp-2022-274

**Anonymous Referee #1**
**Referee comment on "Enhanced sulfur in the UTLS in spring 2020" by Laura Tomsche et al., Atmos. Chem. Phys. Discuss., https://doi.org/10.5194/acp-2022-274-RC1, 2022**
**1) comments from referee**

**This paper by Tomsche et al. presents new SO2 and SO42- (along with other trace gas species) measurements obtained during the BLUESKY aircraft research campaign in May/June 2020, during the European Covid-lockdown which offered an unprecedent opportunity to measure air masses characterised by drastically decreased air pollution. Interestingly, the authors find strongly enhanced SO2 and SO42- concentrations in the upper troposphere and lower stratosphere, respectively, despite overall reduced air pollutant emissions. While these observations are of great value and offer a benchmark against which the chemistry in air quality and other chemistry transport models can be tested, I find that the conclusions provided in the abstract not to be supported by enough evidence within the paper. The explanations for the enhanced sulfur-species concentrations comprise a range of different possibilities, which the authors fail to meaningfully constrain by their evaluations. I therefore cannot recommend this paper for publication in ACP in its current form.**

2) **author's response**

**We thank the reviewer for the comments. We went back to the analysis and followed your comments to improve the manuscript and to provide more evidence on the unique SO$_2$ and SO$_4$ data that we measured during spring 2020. We calculated more specific back trajectories and added more information, e.g. H$_2$O profile and additional tropopause information, to ensure a more in-depth understanding of the whole meteorological situation. We hope that the revised version of the manuscript now more clearly supports the conclusions provided in the abstract In detail, we extended the explanations on the potential sinks and sources for SO$_2$ and set the chemical tropopause in the context of the thermal and dynamical tropopause. Moreover, we discussed air mass origins for elevated SO$_2$ and also SO$_4$ based on the extended trajectory analysis. Finally, we rephrased some of our argument to be easier readable and more comprehensible.**

Specific comments:
**1) comments from referee**

**L72, Section 2.1 BLUESKY mission:**
**It would be interesting for the reader to have a short summary of the mission goals added here. It seems obvious but it should be made more explicit as you have done nicely in the abstract.**

1a) **author's response**

To give the reader a short overview of the goals of the missions, we added a short summary of the goals of the BLUESKY mission in Section 2.1

1b) **manuscript changes**

L77-80: "The period covered the first weeks of the COVID-19 lockdown in Europe and thus offered a unique opportunity to investigate an unprecedented situation of reduced anthropogenic emissions from industry, ground, and airborne transportation. The goal of BLUESKY was to explore the changes in

the atmospheric composition and gain new insights on how anthropogenic emissions perturb chemical and physical processes in the atmosphere."

**2) comments from referee**

**Figure 1. Could the paths of the different flights be represented in different linestyles or shades of blue/red to help emphasize that they were carried out on different days? The current figure may show the coverage, is otherwise though not very informative.**
**Also, you talk about coordination of the flights between Falcon and HALO, but the coverage is rather different. What was the main aim of the coordination?**

2a) **author's response**

We agree with the reviewer that the reader might get the impression that HALO and Falcon flight tracks differ from each other. Despite different aircraft ranges of HALO and Falcon (max altitude of 14.5 km and 12.5 km and around 10 and 4 flight hours, respectively), we succeeded to have 5 out of 8 HALO missions together with HALO and Falcon. Here we focus on 20 flights that were flown with both aircraft over the course of BLUESKY, while the majority of flights were combined. This was made possible due to combined flight planning for HALO and Falcon. During these flights both aircraft probed similar air masses or the same region. In order to account for differences in endurance and flight velocity, We took profit from the different ranges and in some occasion HALO flew above Falcon to extend the range and Falcon performed profiling measurements during refuelling. To make this more clear, we updated Figure 1 and adapted the caption and added text in section 2.1

2b) **manuscript changes**

L81-83: "The payload of both aircraft was complementary to obtain a comprehensive dataset. Especially, five days with coordinated flights over Germany (23 May, 26 may, and 28 May) and over the North Atlantic (30 May, 02 June), the payload offered the opportunity to probe the air masses in more detail."

**Caption L91-92: "…Coordinated flights (Falcon: orange; HALO: cyan) were performed over Germany on 23 May, 26 May, and 28 May and twice (30 May and 02 June) as both aircraft headed towards the North Atlantic, west of Ireland. "**

**3) comments from referee**

**L96 Sensitivity of measurement to moisture: It seems somewhat arbitrary to use a specific altitude as cut-off since you could find samples with high/low moisture content even below/above 8 km depending on the meteorological situation you're flying in. What is the range of H2O mixing ratios you can/cannot easily perform this correction for? Did you measure H2O and if yes, with which instruments?**

3a) **author's response**

**On board Falcon, $H_2O$ was measured with the onboard Lyman-alpha absorption instrument, which is part of the meteorological sensor system.**

**For the BLUESKY $SO_2$ data, the moisture correction could be applied for water vapor mixing ratios roughly up to 500 ppm, which corresponds to an altitude of ca. 7.5 km. With higher water vapor mixing ratios, the uncertainty in the analysis increases. The uncertainty of the correction is around 1.4%, for the low water vapor mixing ratios, for higher water vapor the uncertainty dominates and leads to an increase of the total uncertainty. The uncertainty of the correction is quantified in the uncertainty analysis. Additionally, fast ascends and descends impact the data quality and thus the uncertainty.**

**To limit the impact of both effects, we restricted our analysis to above 8 km as the focus on the presented study is the UTLS region. We adapted the text**

3b) **manuscript changes**

**L105-106:** "…, a correction is more difficult and reduces the data quality. As the focus of the present study is the UTLS region, we limited our analysis on altitudes above 8 km and thus ensure the data quality. "

**L111:** "…The total uncertainty is on average 22.7% for SO2 and included the uncertainty of the moisture correction**.**"

**L115-116:**" Additionally, water vapor ($H_2O$) was measured with the Lyman-alpha absorption instrument integrated in the meteorological sensor system.**"**

**4) comments from referee**

**L138 Could be written more clearly. I suggest to replace 'along the same longitudes and vice versa' with 'and vice versa along the longitudes outside this range'**

4a) **author's response**

I implemented your suggestion.

4b) **manuscript changes**

L150: "… and 28 ppb (HALO) and vice versa along the longitudes outside this range, when CO mixing ratios were enhanced…"

**5) comments from referee**

**L142 Related comment. I suggest to explicitly say that O3 and HNO3 are positively correlated as expected and repeat the longitude range here, since 'in the mentioned longitude range' may not be clear to readers given that you talk about two in L138.**

5a) **author's response**

Following the suggestion, we changed the text

5b) **manuscript changes**

L153-154 to:" Between 6°W and 3°E, $O_3$ and $HNO_3$ are positively correlated, as expected, and $HNO_3$ mixing ratios increase up to 1.6 ppb."

**6) comments from referee**

**L148 I would rewrite this sentence here to point towards the more in-depth analysis and discussion in Section 5 and without claiming it is 'just' from long-range transport. As it currently stands here, I cannot judge from Figure 5 whether the evidence you provide is good enough to underpin this result. For example, I see one trajectory rising from rather low altitude starting around the Eastern coast of North America ending at the measurement location.**

6a) **author's response**

**We agree, that mentioning long range transport in this section is not the right spot and an indication for further explanations later on fits better. Now the last sentence reads:**

6b) **manuscript changes**

**L160:** "An in-depth analysis and discussion on potential explanations for this or similar features will follow in section 4."

**7) comments from referee**

**Figure 3: I suggest adding the tropopause height onto the figure.**

7a) **author's response**

The dynamical tropopause as 2PVU is now added to Figure 3 including updated caption.

7b) **manuscript changes**

L201-202 (caption):" Additionally, the black line at 335 K roughly indicates the dynamical tropopause (as 2 PVU) based on ECMWF/ERA5 analysis along the Falcon flight tracks."

**8) comments from referee**

**L163 It would seem important to indicate the average altitude of the dynamical and/or thermal tropopauses over the region during this time period as well, not to give the impression of choosing what fits best your lower bound of the mixing layer. I would expect the 2PVU tropopause being close to the 330K isentrope, so it would confirm your choice of ozone value for defining the chemical tropopause.**

8a) **author's response**

The dynamical tropopause is based on ECMWF/ERA5 analysis along the Falcon flight tracks. The median theta profile exceeds at 335K the 2 PVU. After adding in Figure 3 the dynamical tropopause height, which confirmed our choice of the chemical tropopause at 120 ppb $O_3$, we added a sentence in the text.

8b) **manuscript changes**

L176-180: "The chemical tropopause is marked by strong gradients in several tracers. O3 as a common indicator shows a kink in its median profile at around 140 ppb and 340 K potential temperature, which is within the limits given by Thouret et al. (2006). The dynamical tropopause, displayed as the 2 PVU based on ECMWF/ERA5 data along the Falcon flight tracks (Fig. 3), is around 335 K and thus has a similar height as the chemical tropopause."

**9) comments from referee**

**L203 Is this an expected concentration range for the mixing layer to be found in at these latitude bands and season?**

9a) **author's response**

**Yes, it is within the range of latitude and season. Pan et al. (2004) and Hoor et al. (2002) showed a similar extension of the transition layer at midlatitudes in summer. I added some references in the text.**

9b) **manuscript changes**

**L219-221:** "The mixing layer almost extends over the whole $O_3$ range from 150 ppb to 400 ppb, similar to other mixing layers in the same latitude and season (Hoor et al., 2002; Pan et al., 2004) and thicker in comparison to a winter polar mixing layer (Fischer et al., 2000)."

**10) comments from referee**

**Figure 4 and discussion section 3.3: It is really hard to follow the discussion of this figure. The only really outstanding feature I can detect when looking at these panels is that there are some high concentrations in SO42- in the troposphere (at ozone values below 100 ppbv and CO values between 100 and 125 ppbv). It would be nice to have this feature explained by the backward trajectories. Otherwise, it is expected that SO2 decreases and SO42- increases as one goes into the stratosphere due to the aging of air, which is reflected both in the very strong anticorrelation between the two sulfur species and also in the SO42- / (SO42- + SO2) ratio visible in Figure 2 and 3. Maybe you could circle the points you are referring to in the figure in case I have missed what you are really referring to?**

10a) **author's response**

Thanks for pointing to the feature. The outstanding feature in the $SO_4^{2-}$, you mentioned, is out of the scope of this study and should not be included in the data. In the updated Figure, it is removed to not confuse the reader. Additionally, the outlier mentioned in L224-225 is now circled in Fig. 4c. Figure 4 is updated including the caption.

10b) **manuscript changes**

**L245-248 (caption): "Figure 4: Tracer-tracer correlation for a) 30 sec data CO - $O_3$ in black for whole altitude range and for heights above 8 km $SO_4^{2-}$ is color-coded from HALO flights, b) CO - $O_3$ in black for whole altitude range, and with color-coded $SO_2$ (above 8 km) from Falcon flights when $O_3$ was available, and c) 1 sec data CO - $HNO_3$ with color-coded $SO_2$ from Falcon flights. In c) a grey circle marks an outlier with high $SO_2$, CO and $HNO_3$."**

L231-232: "One $SO_2$ outlier with higher mixing ratios at enhanced $HNO_3$ und reduced CO can be identified (Fig 4c; grey circle)."

L241-242: "The back trajectories of $SO_2$ and $SO_4^{2-}$ cases support the assumption that the origins differ, especially concerning the different altitude ranges of the trajectories (Fig. 5)."

**11) comments from referee**

**L215/Figure 5. This evaluation would be more effective and convincing if you would identify the exact times/altitude/geographical location of when unusual trace gas observations are made and then calculate your backward trajectories from locations on a finer grid point around these. I then would suggest not only to show lat-lon plots, but also time-altitude evolution of the trajectories and also PV along with it, so that the reader can see where/when tropospheric influence or strong uplift within the Asian monsoon may have happened.**

11a) **author's response**

In the new Figure 5, exact time/altitudes/geographical location and also $SO_2$ or $SO_4$ mixing ratios are included. **It includes lat-lon, as well as hour before release-altitude plots. HYSPLIT has no tropopause height information, thus the closest indication is the potential temperature, which is given as color-coded tracks. Text and caption are adapted.**

11b) **manuscript changes**

**L234-236: "…** long range transport in the UTLS region for this case (Fig. 5a-b). Generally, the trajectories do not indicate transport from local PBL sources for cases with elevated $SO_2$. In Figure 5a-h, examples for cases with elevated $SO_2$ are shown, including the case on 02 June 2020 described in section 3.1 (Fig. 5g/h)."

**L253-258: "Figure 5: HYSPLIT 360 hours back trajectories calculated for cases with elevated $SO_2$ (a-h; Falcon) and $SO_4^{2-}$ (i-p; HALO) mixing ratios. The release points started in the vicinity of these events. In the left column: in black are the flight tracks, color-coded is the potential temperature along the trajectories to indicate the transport altitude. The right column represents the trajectories as hours before release vs. altitude, also with color-coded potential temperature The cases of enhanced $SO_2$ were on 19 May 2020 (a, b), 23 May 2020 (c, d), 30 May 2020 (e,f), and 02 June 2020 (g, h). The cases of elevated $SO_4^{2-}$ were on 23 May 2020 (i, j), 26 May 2020 (k, l), 30 May 2020 (m, n) and 02 June 2020 (o, p)."**

**12) comments from referee**

**Figure 5 Why is there an empty panel (j)?**

12a) **author's response**

Figure 5 is replaced by a new one including other trajectories.

**13) comments from referee**

**L253-262 I cannot fully follow your argumentation here. Don't you miss to account for the 72% reduction in air traffic due to Covid when you calculate your 50% increase in aviation SO2 emission increases?**

13a) **author's response**

I see your point and formulated the argumentation more precisely. The 50% increase is not related to the 72% reduction, but is a theoretical value for a scenario without COVID19 restrictions. I adapted the paragraph

13b) **manuscript changes**

L282-287: "…by a factor of 1.5 of the 2010 air traffic for a scenario without COVID19 restrictions in 2020 and consequently a theoretical increase of 50 percent in aviation $SO_2$ emissions for 2020, given that the sulfur content of the kerosene is unchanged (Lee et al., 2021; Miller et al., 2009). In 2020, The remaining air traffic of 28% (in comparison to 2019) corresponds to roughly 40% of the 2010 air traffic and might hence in part explain the $SO_2$ mixing rations detected in the upper troposphere, which are still higher than remote background measurements (Williamson et al., 2021)."

**14) comments from referee**

**L291-295 This conclusion is not very satisfying. Much more should have been done to better disentangle all the different possible explanations of the enhanced SO2 throughout the upper troposphere. The reader knows in principle as much as before reading your paper about the potential sources of the enhanced sulfur species. If you could do the backward trajectories in a more targeted way, that is identifying the origin of high-SO2 events for different flights more meaningfully, this could be a much better step to answer the question you wish to answer.**

14a) **author's response**

**We added a H₂O profile in Fig. 3g as suggested by the 2ⁿᵈ reviewer, to stress the explanation of the SO₂ sinks and added for the H2O measurement Martin Zöger as coauthor. Further, we calculated more specific trajectories for high SO₂ and also high SO₄ cases and replaced Fig. 5 with a new one, which includes representative examples of back trajectories. We adapted the text with more in-depth analysis regarding trajectories in section 4, especially we rewrote the paragraphs about SO₂ sinks, SO₂ sources, and the conclusion including new trajectory analysis.**

14b) **manuscript changes**

**L137-140:** "For selected cases with either elevated $SO_2$ or $SO_4^{2-}$ mixing ratios, 360 h back trajectory ensembles were calculated. One ensemble consists of 27 single trajectories, which are offset by one meteorological grid point in the horizontal and 0.01 sigma in the vertical coordinate."

**L 296-319:** "In addition, further sources could have contributed to the SO2 budget in the upper troposphere. To analyse the origin of air masses with elevated SO2, HYSPLIT back trajectories are calculated and representative examples are plotted in Fig. 5a-h. SO2 emissions from anthropogenic and natural sources in the PBL can be lifted to the UT via convection or via warm conveyor belts and transported to the measurement region. Arnold et al. (1997) reported an extended layer of enhanced SO2 with maxima of up to 3 ppb in the Northeast Atlantic, which was an air mass uplifted and transported from the polluted PBL from the eastern United States. A few cases show trajectories with similar pathways, like the example in Fig. 5e/f. Nevertheless, the PBL contacts are also over the Pacific and East Asia. The latter one suits better to the findings of Fiedler et al. (2009) who observed the uplift of polluted air masses from East Asia via warm conveyor belts and upper tropospheric long-range transport towards Europe. Further, the Asian monsoon also serves as a vertical transport pathway for emissions from the PBL up to high altitudes, where the air mass can enter the LS and horizontally be transported either eastwards (Vogel et al., 2014, 2016) or can be horizontally transported in the UT (Tomsche et al., 2019) and finally reach Europe. Similar trajectory pathways can be found for the cases in Fig. 5a/b and 5g/h. These trajectories indicate long range transport in the UT and could have been impacted by the Asian monsoon. Generally, the trajectories with elevated SO2 (Fig. 5a-h) show lower potential temperatures in comparison to the trajectories calculated for elevated SO42- (Fig. 5i-p). Hence, long range transport of SO2 enriched PBL air masses could have contributed to the observed BLUESKY SO2 mixing ratios in the UT. In contrast, the trajectories do not indicate local transport from the central European PBL to the UT, hence the transport of SO2 from wildfires in Germany in May 2020 (European Commission, 2021) to the UT seems negligible. Even if the transport of the smoke was via self-lofting (Ohneiser et al., 2021), i.e. absorption of sunlight leads to warming of the ambient air and thus lifting of the smoke, the transport is slow and so SO2 might already been transformed to SO42- before reaching the UTLS and does not contribute to the elevated SO2 in the UT. Moreover, the decrease of SO2 in the LS, as expected, does not support transport of SO2 beyond the UT into the LS neither via convection nor warm conveyor belts. This can be confirmed by the trajectories for SO42- as they indicate long range transport at high altitudes with negligible influence from lower altitudes. The trajectories helped identifying potential source regions of SO2."

**L320-333:** "Beside the sources, also sinks of $SO_2$ can alter the $SO_2$ concentrations in the UTLS. Beside the conversion to $H_2SO_4$, leading to sulfate particles, $SO_2$ is removed from the atmosphere by wet and dry deposition. $SO_2$ can be scavenged by clouds, which lead to a significant reduction of the $SO_2$ lifetime (Lelieveld, 1993). Van Heerwaarden et al. (2021) investigated the meteorological situation in spring 2020 and found that a stable high pressure system over Europe lead to a lower cloud fraction in comparison to the mean 2010-2019 period over Europe. This would lead to less cloud processing and reduce $SO_2$ sinks. Furthermore, elevated humidity favours the faster conversion of $SO_2$ to $SO_3$ and sulfate, as water vapour enhances the potential for elevated OH concentrations (Pandis and Seinfeld, 2006). As reported by Schumann et al. (2021a, 2021b) the UTLS was drier in spring 2020 in Europe in comparison to previous years. The median $H_2O$ profiles during BLUESKY reach only

mixing ratios of up to 268 ppb in the upper troposphere (Fig 3g), which is still within the range of typical springtime $H_2O$ mixing ratios in the upper troposphere (e.g. Hegglin et al., 2009; Kaufmann et al., 2018). But van Heerwaarden et al. (2021) found with respect to humidity and cloud cover, spring 2020 was amongst the springs with the lowest values. . Thus, the lower available $H_2O$ led to lower OH concentrations during BLUESKY period, which implies less chemical processing and hence a reduction of $SO_2$ sinks. Less $SO_2$ sinks could lead to an enhanced $SO_2$ lifetime in the UTLS and thus higher $SO_2$ mixing ratios. "

**L334-338:** "In sum, the enhanced $SO_2$ mixing ratios at cruise levels in Europe in spring 2020 can possibly be explained by a non-negligible aviation $SO_2$ contribution, WCB or convective transport from the boundary layer, followed by long range transport, and the prolonged $SO_2$ lifetime caused by the unusually dry UTLS conditions. Neither the sources nor the sinks could separately explain the $SO_2$ mixing ratios in the UTLS. Beyond that, we are not able to analyse in more detail the different amounts of the aforementioned factors and how they contribute to single flights. "

**15) comments from referee**

**Section 5: It would be interesting to see whether the clear anti-correlation in SO42- with ozone seen for the flight in Figure 2 is also found for other flights. This could then be used to identify truly anomalous correlations for which again more targeted backward trajectories could potentially yield much more insight than is provided here. As the discussion is presented here, multiple explanations from existing literature but no additional insights are provided, leaving the reader once more unsatisfied about having learned anything new.**

15a) **author's response**

I suppose that the referee means the correlation between $O_3$ and $SO_4$. $SO_4$ is clearly correlated with $O_3$ for all flights in the UTLS region presented here, comparable to the flight presented in Figure 2. In order to provide further insight in the airmass origins of cases of elevated $SO_4$, we calculated specific trajectories. In Figure 5i-p four representative cases are displayed.

15b) **manuscript changes**

**L253-258 (caption): "Figure 5: HYSPLIT 360 hours back trajectories calculated for cases with elevated $SO_2$ (a-h; Falcon) and $SO_4^{2-}$ (i-p; HALO) mixing ratios. The release points started in the vicinity of these events. In the left column: in black are the flight tracks, color-coded is the potential temperature along the trajectories to indicate the transport altitude. The right column represents the trajectories as hours before release vs. altitude, also with color-coded potential temperature The cases of enhanced $SO_2$ were on 19 May 2020 (a, b), 23 May 2020 (c, d), 30 May 2020 (e,f), and 02 June 2020 (g, h). The cases of elevated $SO_4^{2-}$ were on 23 May 2020 (i, j), 26 May 2020 (k, l), 30 May 2020 (m, n) and 02 June 2020 (o, p)."**

L347-348: "In Figure 4a, this layer represents the mixed layer. The $SO_4^{2-}$ correlates well with $O_3$ for all flights, similar to the flight on 02 June 2020, presented in section 3.1 (Fig. 2)."

L368-374: "As mentioned above, we calculated HYSPLIT back trajectories for cases of elevated SO42-. In Figure 5i-p, representative examples of trajectories are displayed. The majority indicates long range transport at high altitudes with potential temperatures between 343 K and 465 K. While the lower range of the potential temperature is associated with midlatitude tropopause height, the upper values are clearly associated with the stratosphere. As only few trajectories indicate lower potential temperatures, we assume that the majority of elevated SO42- is already in the stratosphere 360 h before sampling and hence influence from the troposphere is negligible."

**16) comments from referee**

**L360 I do not think that you have provided real evidence that you have measured enhanced SO42- concentrations in the stratosphere, since we do not know what the 'normal' background would be. The difference seen in Figure 6 could simply be due to measuring within the troposphere at altitudes 11-15 km for the CAFE-Africa campaign data, which to my understanding was performed within the tropics.**

16a) **author's response**

Yes, CAFE-Africa was performed in the tropics, but we only used a subset of flights north of 38°N as described in L354 -356 to be able to compare to BLUESKY. For a better comparability of stratospheric air masses, I added the tropopause height for BLUESKY and CAFE-Africa in Fig. 6. Further, we also compared the $SO_4$ mixing ratios with studies of volcanic quiescent periods and other available studies, to receive a bigger picture about the situation in spring 2020 in comparison to other situations (e.g. Martinson et al., 2005 in L396).

I adapted the text. Moreover, the new trajectory calculation, as mentioned above, provide more insight in the origin of the elevated $SO_4$. Altogether it improved our analysis and helped to provide more evidence of the measured $SO_4$.

Changes in the text are given in the previous comment/answer section concerning trajectories with respect to elevated $SO_4$ cases, and additional changes in the text are the following:

16b) **manuscript changes**

**L341-342:"**. This altitude reflects also the thermal tropopause height during BLUESKY (Fig. 6b)."

**L363: "**For the CAFE-Africa subset the thermal tropopause was slightly higher than during BLUESKY (Fig. 6b)."

L412-415: "But it seems unlikely, as the trajectories for elevated $SO_4^{2-}$ (Fig. 5i-p) stay at high altitudes (and high potential temperatures) and thus mixing from the troposphere into the stratosphere is negligible. A few trajectories indicate a downward transport from higher altitudes, thus an origin deeper in the stratosphere (Fig. 5k/l), which could be a hint for OCS as $SO_4^{2-}$ precursor."

**L429-430:** "Back trajectories provided indications of other boundary layer $SO_2$ sources from convective or WCB transport and further long-range transport in the UT, which could have contributed to a small extend."

And L438-440: "… primary sources of the enhanced stratospheric $SO_4^{2-}$ concentrations measured during BLUESKY, because back trajectories mainly showed long range transport in the lower stratosphere. "

*Minor corrections/typos:*
**17**) **comments from referee**

**L31 geo engineering --> geoengineering**

17a) **author's response**

 **Done L32**
**18**) **comments from referee**

**L90 in flight --> in-flight**

18a) **author's response**

 **Done L98**
**19**) **comments from referee**

**L146 delete 'to' in front of '266 ppb…'**

19a) **author's response**

 **Done L158**

**20) comments from referee**

**L179 write either 'with stratospheric sulfate aerosol' or 'with the stratospheric sulfate aerosol layer'**

20a) **author's response**

**L196: changed to 'with stratospheric sulfate aerosol'**

**Reviewer 2 (Daniele Visioni):**

**21) comments from referee**

Referee comment on "Enhanced sulfur in the UTLS in spring 2020" by Laura Tomsche et al., Atmos. Chem. Phys. Discuss., https://doi.org/10.5194/acp-2022-274-RC2, 2022

This manuscript is overall an excellent overview of the BLUESKY mission, and presents some rather interesting measurements of the UTLS in a peculiar case (the beginning of the COVID-driven decrease in emission). It has the potential to be a rather important reference going forward, but clearly the manuscript needs more work, and to be cleaned up a bit.

21a) **author's response**

Thanks for writing a review. I spend more time analyzing data, especially back trajectories to improve the manuscript and I rephrased some paragraphs to be more precisely in the explanations.

**22) comments from referee**
I generally agree with all the comments already laid out by Reviewer 1, so I find it pointless to repeat them one by one: but in particular, I found the possible causes of enhanced SOA,, to be a bit too much handwaved in section 4.

For instance, at L. 283, the paragraph starts with "Beside the conversion to H2SO4..." but then continues by mentioning that most of the SOA,, would be reduced by its conversion to sulfates through OH oxidation. Something you clearly go back to at the end of the paragraph, by simply repeating the same thing. But no convincing proofs are given. Even just the addition of a vertical profile of humidity from ERA5 reanalysis (which is what van Heerwaarden et al. uses) would go a long way into proving that, indeed, SOA, lifetime has been increased by lower OH concentrations.

22a) **author's response**

Thank you for the idea of a $H_2O$ profile. I decided to use in situ data onboard HALO and Falcon in Fig. 3g. OH was also measured during BLUESKY onboard HALO, but the data are not finalized yet and are not available for the present work. Nevertheless, the OH trend supports our interpretation. Added text and rewrote the paragraph about the $SO_2$ sinks.

22b) **manuscript changes**

L 115-116: "Additionally, water vapor ($H_2O$) was measured with the Lyman-alpha absorption instrument integrated in the meteorological sensor system."

L 130-131: "Water vapor was measured with the tunable diode laser (TDL) hygrometer SHARC (Sophisticated Hygrometer for Atmospheric ResearCh)."

L182-184 "The profiles of the tropospheric tracer $H_2O$ also decrease with height, following the CO profile, but with a less pronounced step around the chemical tropopause. The $H_2O$ mixing ratios are 26-268 ppb (Falcon) and 3-153 ppb (HALO)."

L320-333:" Beside the sources, also sinks of $SO_2$ can alter the $SO_2$ concentrations in the UTLS. Beside the conversion to $H_2SO_4$, leading to sulfate particles, $SO_2$ is removed from the atmosphere by wet and dry deposition. $SO_2$ can be scavenged by clouds, which lead to a significant reduction of the $SO_2$ lifetime (Lelieveld, 1993). Van Heerwaarden et al. (2021) investigated the meteorological situation in spring 2020 and found that a stable high pressure system over Europe lead to a lower cloud fraction in comparison to the mean 2010-2019 period over Europe. This would lead to less cloud processing and reduce $SO_2$ sinks. Furthermore, elevated humidity favours the faster conversion of $SO_2$ to $SO_3$ and sulfate, as water vapour enhances the potential for elevated OH concentrations (Pandis and Seinfeld, 2006). As reported by Schumann et al. (2021a, 2021b) the UTLS was drier in spring 2020 in Europe in comparison to previous years. The median $H_2O$ profiles during BLUESKY reach only mixing ratios of up to 268 ppb in the upper troposphere (Fig. 3g), which is still within the range of typical springtime $H_2O$ mixing ratios in the upper troposphere (e.g. Hegglin et al., 2009; Kaufmann et al., 2018). But van Heerwaarden et al. (2021) found that the humidity and the cloud cover of spring 2020 were amongst the springs with the lowest values. Thus, the available $H_2O$ lead to lower OH concentrations during BLUESKY period and would imply less chemical processing and hence would also reduce $SO_2$ sinks. Less $SO_2$ sinks could lead to an enhanced $SO_2$ lifetime in the UTLS and thus higher $SO_2$ mixing ratios. "

**23) comments from referee**
Something similar can be said for Section 5.

A rather long list is given, but I don't find a compelling critical assessment of possible causes. Starting from the bottom, sure, we know COS contributes to the stratospheric aerosol layer, but what causes would have produced an increase significant enough to be observed in this campaign (noting that photolysis of OCS happens much higher up, so its eventual conversion to aerosols and signal at the lower altitudes would not be as instantaneous as for SOA). So in all of this, Figure 6 is really not that helpful (also, the legend in panel b) seems to be hiding the 2017 data...). Are we sure those different measurements are performed over similar regions? The Andrés Hernández (2022) reference seems to be targeted at lower altitudes, and I can't find SO4 measurements there, so it's hard to judge if the comparison is correct or not - and it's the only point offered for comparison! This hand-waviness is reproduced as is in the Conclusions, where the phrase " The enhanced stratospheric sulfate aerosol, which was observed, was likely impacted by the volcano Raikoke, and smaller sources" doesn't really say anything at all.

23a) **author's response**

Andrés Hernández et al. (2022) presented the EMeRGe-EU campaign in summer 2017 with measurements over Central Europe, even though the measurements are limited to an altitude of 10 km, the data provide an additional comparison for the $SO_4$ profile in the upper troposphere. In the UT, their $SO_4$ mixing ratios agree with the BLUESKY data well. The European subset of the CAFE-Africa data

set is another comparison (which has not been published before) and covers the same altitude range than the BLUESKY mission, thus offering the option to compare UT and LS $SO_4$ mixing ratios.

Figure 6 is updated and the legend is not hiding any data. Further, I included for the BLUESKY Falcon flights (a) and HALO flights (b) a dynamical tropopause (~2 PVU) and a thermal tropopause, respectively. In b) also the thermal tropopause for the CAFE-Africa subset is included. Further comparison to other studies is given in the text and Fig 6b) (e.g. Schmale et al., 2010). Moreover, an additional potential source, which could contribute to elevated SO4 is added in the text:

23b) **manuscript changes**

L418-421: "Further, the Junge layer might also influence the $SO_4^{2-}$ mixing ratios in the LS. Even though the Junge layer is most pronounced at higher altitudes (Junge et al., 1961), it could extend further down or due to the downward transport mentioned above (Fig. 5k/l) $SO_4^{2-}$ could be transported downward from the Junge layer into our measurement altitudes and thus contribute to the elevated $SO_4^{2-}$."

**24) comments from referee**
In a future revision, both Section 4 and 5 need to be strengthened and more analyses need to be performed to make sure this really reads like a Research Article. Otherwise, the editor and the authors could consider shortening those parts and limiting the scope of the work to a "Measurement report" type of manuscript for ACP.

24a) **author's response**

I followed your suggestion and performed more analysis, especially more trajectory analysis, to get a better understanding on the air mass origins of elevated $SO_2$ and also $SO_4$ cases. Further, I included $H_2O$ profiles to strengthen the assumption about the exceptional dry weather situation in the UTLS as mentioned in other studies. I think the manuscript improved to present a research article rather than a measurement report.

**25) comments from referee**

One more typo: "One major source of $SO_4^{2-}$ in the stratosphere are volcanic eruptions" should be "is" and not "are"

25a) **author's response**

Done L374

**Anonymous Referee #3**
**Referee comment on "Enhanced sulfur in the UTLS in spring 2020" by Laura Tomsche et al., Atmos. Chem. Phys. Discuss., https://doi.org/10.5194/acp-2022-274-RC1, 2022**

**1) comments from referee**

This manuscript provides an overview of the BLUESKY mission during the 2020 COVID-19 lockdown in Europe. The authors present interesting measurement results of $SO_2$ and $SO_4^{2-}$, along with other trace gases and aerosols, at UTLS, which have the potential to be an important reference to future chemistry and modeling development. The authors have revised the manuscript with more discussions based on the comments from previous reviewers.

In addition to the comments from previous reviewers, I'm particularly interested in how wildfires contribute to $SO_2$/$SO_4^{2-}$ profile changes. Although the authors cited previous studies that demonstrated smaller amount of $SO_2$ released from wildfires compared to that from volcanic eruptions, this wouldn't necessarily represent the BLUESKY case since the meteorological conditions may be different between 2019 and 2020. Based on the 2020 EU JRC wildfire report (https://ec.europa.eu/commission/presscorner/detail/en/ip_21_5627), there were still significant wildfire events in Europe, especially in Germany, during May 2020. If those wildfires were intense enough, $SO_2$ and $SO_4^{2-}$ at UTLS could be influenced. The authors should provide sufficient evidence to convince readers how these "local" wildfires, in addition to long-range transport $SO_2$ from other continents, didn't contribute significantly to the $SO_2$ and $SO_4^{2-}$ profile changes during the BLUESKY mission. Additional discussion about wildfires can help improve the manuscript.

1a) **author's response**

We thank the reviewer for the comments. We revised the manuscript and added details on wildfires and how they contribute or not to the mixing ratios of SO2 and SO42- in the UTLS. In particular, we discussed that "local", i.e. European, fires had probably very low impacts on the May/June 2020 UTLS. Explanations are in section 4 and 5 for SO2 and SO2-, respectively.

1b) **manuscript changes**

L312-316: "In contrast, the trajectories do not indicate local transport from the central European PBL to the UT, hence the transport of $SO_2$ from wildfires in Germany in May 2020 (European Commission, 2021) to the UT seems negligible. Even if the transport of the smoke was via self-lofting (Ohneiser et al., 2021), i.e. absorption of sunlight leads to warming of the ambient air and thus lifting of the smoke, the transport is slow and so $SO_2$ might already been transformed to $SO_4^{2-}$ before reaching the UTLS and does not contribute to the elevated $SO_2$ in the UT. Moreover, …"

L388-393: "Ohneiser et al. (2021) discussed self-lofting as a potential transport pathway in the UTLS for these Siberian fires in the absence of strong vertical motion in July 2019. The smoke plume could raise and reach the UT and further ascent into the LS. During the slow ascent, the emissions alter chemically, in the case of $SO_2$, it is transformed to $SO_4^{2-}$. Finally, the $SO_4^{2-}$ could have contributed to the enhanced $SO_4^{2-}$ in the LS. Further, wildfires in central Europe in May 2020 (European Commission, 2021) could also have undergone this self-lofting process as the trajectories do not indicate uplift over Europe and thus might additionally have contributed to elevated $SO_4^{2-}$ in the UTLS."

**Minor comments:**

**2) comments from referee**

Ln 22, 124, 125, 128, 353, 358, 359. Keep the number expression consistent. A thin space is suggested before and after the plus-minus sign.

2a) **author's response**

-> done L22, L12, L125, L127, L359, L364, L365

**3) comments from referee**

Ln 55. "(46°N, 0.8 Tg SO2)" is suggested.

3a) **author's response**

-> done L56

**4) comments from referee**
Ln 59. Typo: "important source of stratospheric aerosol are intense wildfires, ..." should be "is" instead of "are."

4a) **author's response**

-> done L59
**5) comments from referee**

Fig. 1 The bottom axis at 0 longitude seems to have an additional character embedded.

5a) **author's response**

-> updated Figure 1
**6) comments from referee**

Ln 64. You may want to spell out COVID at its first appearance in the text. Also, be consistent with using COVID-19 or COVID19 in the manuscript.

6a) **author's response**

-> done L63 and used COVID-19 (L280, L283)
**7) comments from referee**

Ln 128. Lower case of N for "nitrous."

7a) **author's response**

-> done L128
**8) comments from referee**

Ln 132. This sentence should be combined with the previous sentence.

8a) **author's response**

 -> done L130

**9) comments from referee**
Ln 137. A proper citation/reference to the GDAS dataset is mandatory.

9a) **author's response**

-> done L137
**10) comments from referee**

Ln 159. May replace "will follow" with "is presented."

10a) **author's response**

-> done L160
**11) comments from referee**

Fig.2 caption. "… Plotted are a) $SO_2$, …, and f) altitude across longitudes ##°W–##°E."

11a) **author's response**

-> done L164/165
**12) comments from referee**

Fig. 3 caption. Superscripts for 25th and 75th, be consistent with how they are used in the main text.

12a) **author's response**

-> done L199
**13) comments from referee**

Ln 205. "… 310 K and 340 K; above the chemical tropopause the sum increases up to …"

13a) **author's response**

-> done L207
**14) comments from referee**

Ln 395. Typo: "were" instead of "where."

14a) **author's response**

-> done L386